# Integrated approach to model distribution and assess habitat suitability of killifish species in Oman's local streams (wadis) under current and future climate conditions

**Aziza S. Al Adhoobi**[1]*, **Amna Al Ruheili**[2], **Saud M. Al Jufaili**[1]*, **Wenresti Gallardo**[1]

**1** Department of Marine Science and Fisheries, College of Agricultural and Marine Sciences, Sultan Qaboos University, Muscat, Oman, **2** Department of Plant Science, College of Agricultural and Marine Sciences, Sultan Qaboos University, Muscat, Oman

☯ These authors contributed equally to this work.
* a.aladhoobi@squ.edu.om, azizco83@gmail.com (AA); sjufaily@squ.edu.om (SJ)

## Abstract

Freshwater ecosystems in arid regions possess extraordinary levels of biodiversity, yet they are subject to unprecedented pressures of climate change and anthropogenic activities. We employed an integrated approach of incorporating species distribution modeling (MaxEnt), habitat suitability modeling, and protected area analysis to assess conservation requirements for two endemic/native killifish (*Aphaniops kruppi* and *A. stoliczkanus*) in Oman's freshwater ecosystems. Using MaxEnt with CHELSA bioclimatic variables and topographic indices, we modelled climate change impacts under three shared socio-economic pathways (SSP1–2.6, SSP3–7.0, and SSP5–8.5) spanning 2011–2100. Predictive models demonstrated remarkable accuracy (AUC: 0.974 for *A. kruppi*, 0.950 for *A. stoliczkanus*) revealing unique biogeographical patterns. *A. kruppi* showed restricted southern distribution dependent on monsoon moisture levels, with mean monthly climate moisture index (Cmi_m; 39.9%), mean diurnal range (Bio2; 18.3%), and sediment transport index (STI; 8.4%) as key variables. The distribution of *A. stoliczkanus* exhibited a more expansive northern range influenced by winter precipitation patterns, with precipitation of the coldest quarter (Bio19; 31%), the sediment transport index (STI; 20.2%), and the stream power index (SPI; 13.3%) as key drivers. Climate projections revealed high extrapolation risk (85–95%) with anticipated habitat reductions. Habitat suitability assessment of 12 stream sites (Boyce Index: 0.894) revealed unexpected specialization-dominance trade-off, where optimal *Aphaniops* conditions led to competitive exclusion, resulting in negative correlations between habitat suitability and aquatic biodiversity (Shannon diversity: r=−0.577, p=0.049). Dissolved oxygen emerged as most critical parameter (mean suitability: 0.771±0.308), with only 25% of sites demonstrating a Highly Suitable status. Spatial analysis revealed significant protection gaps: only 0.31–1.34% of high-suitability habitats and 2.6% of high-density wadis currently protected, requiring

**Data availability statement:** All relevant data are within the paper and its Supporting Information files.

**Funding:** "This research was partially funded by the Sultan Qaboos University Financial Support for PhD Students Research Projects (awarded to AA) and by Sultan Qaboos University under the project number IG/AGR/FISH/22/01 (awarded to SJ). The funders had no role in study design, data collection and analysis, decision to publish, or preparation of the manuscript".

**Competing interests:** The authors have declared that no competing interests exist.

6–24-fold increases to meet conservation targets. Species hybridization necessitates landscape-level conservation maintaining connectivity for gene flow. Results demonstrate that desert aquatic fauna conservation requires integrated strategies addressing climate dependencies, multi-habitat corridor protection, tiered water quality standards, and adaptive management accounting for hybridization zones, providing a replicable model for conservation in water-limited environments.

## 1. Introduction

Freshwater ecosystems in arid regions represent some of the most vulnerable biodiversity hotspots globally, with endemic species persisting in isolated refugia amidst increasingly challenging environmental conditions [1–4]. Though freshwater ecosystems comprise less than 1% of the Earth's surface, they contain about 10% of all known species, including many endemic species that have developed unique adaptations to survive in isolated environments [5–7]. Because of their limited and relatively small extent in water-limited landscapes, freshwater ecosystems serve as unique biodiversity refugia that support disproportionately high levels of endemism [2,4,8,9]. The Arabian Peninsula serves as an example for such conservation issues, as it contains remarkable freshwater biodiversity in extremely water-limited landscape types, and endemic fish communities that have developed evolve unique adaptations needed to persist in fragmented wadi systems and spring-fed streams [10–13]. Typical of these endemic/native fish communities, they exhibit specialized traits to adapt to survive in fragmented wadi systems and spring-fed streams [14,15], but together they showcase some remarkable evolutionary ability to persist under harsh environmental conditions [16,17].

Oman is a West Asian country that occupies a position at about 21°00′N 57°00′E along the southeastern edge of the Arabian Peninsula and stretches across roughly 309,500 km² [18–21]. Oman features diverse geographical features consisting of valleys along with deserts mountain ranges and coastal plains [18,22]. Additionally, Multiple regions across Oman experience a variety of climatic conditions from hyper-arid zones with less than 100 mm rainfall to arid zones with 100–250 mm and semi-arid zones with 250–500 mm annual rainfall [23]. Moreover, Rainfall patterns vary considerably across Oman's regions, were the northern and central regions of Oman receive winter precipitation between November and April conversely the southern Dhofar region is affected by a summer monsoon from June to September [24,25]. Besides, Oman's annual rainfall amounts to 31 mm in desert interiors while exceeding 300 mm in mountainous northern regions and reaches around 105 mm throughout southern Oman [24]. Freshwater habitats in Oman contain some of the highest concentrations of freshwater biodiversity on the Arabian Peninsula with 23 species present accounting for 47.9% of native fish species, highlighting Oman's importance as a refuge for endemic species evolved to be highly specialized in their ecology [10,11,26]. The endemic fauna that we find in Oman are adapted to various aquatic environments from permanent springs to ephemeral wadis. The order

Cypriniformes is the most diverse order comprising nine species, followed by Gobiiformes (6 species), and Cyprino-dontiformes (3 species) demonstrating considerable taxonomic diversity in very small aquatic communities [10]. Recent taxonomic research has created a wave of new knowledge about this taxonomic diversity, with multiple new species descriptions made, illustrating the evolutionary significance of these geographically isolated systems [12,27,28]. Killifish of the genus *Aphaniops*, especially the newly described *A. kruppi* and *A. stoliczkanus*, provide an example of evolutionary adaptations to arid conditions, showing both distinct biogeographic patterns and specific ecological requirements related to spring-fed and wadi environments [26,27,29]. Diversity of freshwater fishes in the Arabian Peninsula are shaped by the interaction of natural influences (geographic features, climate, environmental factors) and anthropogenic influences, resulting in fishes occupying highly restricted distributions and specialized adaptations that are considered important components of regional biodiversity and reflect the specialized nature of freshwater ecosystems of the Arabian Peninsula [11,12].

However, these systems are now being subjected to unprecedented anthropogenic and environmental pressures in a water-stressed region from ongoing climate change (global warming, droughts, etc.), continuing human development (e.g., dam construction, water extraction, urbanization, coastal developments, etc.), pollution, population, fragmentation, and introductions of non-native fish species [11,12]. These multiple anthropogenic and environmental pressures pose a serious risk to the unique biodiversity associated with freshwater ecosystems in the Arabian Peninsula that have evolved ecological specialization over time but are now facing increasing environmental challenges that threaten their persistence.

The conservation challenges facing endemic freshwater species are compounded in Oman due to multiple interacting stressors [11]. Climate change clearly poses a significant threat; climate change predictions suggested that many endemic species would lose suitable habitat as precipitation and temperature patterns changed [30–32]. Observational data suggest the region is receiving an annual temperature increase of 0.63 °C/decade with an annual precipitation decrease of 6.3 mm/decade and future projections expected an annual temperature increase of 1.8–2.7 °C under high emission scenario [32]. Increased temperatures and modified precipitation patterns will result in an increase in the frequency of droughts leading to negative impacts on water quality and habitat [33]. Many freshwater species have limited dispersal ability and can be very sensitive to environmental change that can shift food webs and community structure [34]. Oman faces significant extreme weather threats, such as cyclone Gonu in 2007– the most extreme weather event in Oman's history– that delivered a total rainfall of 610 mm [35] and established a single-day rainfall record with 943 mm [36]. The country remains vulnerable to devastating tropical storms as shown by subsequent cyclones for instance Phet (2010), Nilofar (2014), Mekunu (2018), and Shaheen (2021) [36].

Even with a critical need for conservation and an understanding of climate change, there are still substantial knowledge deficiencies associated with understanding species-environment relationships in arid systems [37,38]. Conservation of any species requires addressing unique ecological relationships and interacting threats, with the need for regular monitoring of species and proper management [30,39]. While the use of species distribution models (SDMs) has increased to predict both climate and habitat suitability for freshwater species in general, few species have been able to utilize predictive modeling connected with field based assessments of habitat use in arid regions [40]. Although generally, species distribution modeling methods are highly applicable for broad spatial scales associated with conservation planning, these models often do not properly consider the comparatively fine-scale habitat requirements and micro-habitat choices that determine the persistence of species in water-limited regions [41]. Using habitat suitability modeling to identify areas of refuge for freshwater fish in the area is advancing, where studies of species such as *Cyprinion muscatense*, suggest MaxEnt may provide understanding for climate change effects [42]. Moreover, investigations of *Garra shamal* in the Omani Hajar Mountains identified geomorphological and precipitation patterns that help to determine suitable habitats for endemic fish species [31]. Additionally, wider scale regional assessments have emerged, as conducting a climate change impact analysis to include multiple species endemic to

the freshwater fish fauna of the Arabian Peninsula showed species-specific vulnerability projecting a habitat loss for up to 7 of 9 species of endemic fish under future warming scenarios displaying possible range shift that highlight a broader priority for conservation action in this area of aquatic ecosystem conservation [30]. However, most studies have typically focused on single species or limited geographic areas without examining the broader implications for aquatic ecosystem conservation [43,44].

Previous studies found that freshwater fish in arid regions vary in their specializations to habitats. Some species in the stable environments show narrow tolerance ranges of water temperature, dissolved oxygen, and salinity, while others in variable arid habitat evolved to extreme conditions and exhibit broad tolerance ranges [45,46]. Documented research into species of killifish in Iran has accessed taxonomic diversity and phylogenetic relationships in the genus *Aphanius*, providing important baseline for conservation planning [47,48]. A particularly critical knowledge gap is the relationship between habitat for species, and biodiversity patterns in the ecosystem as a whole, particularly in systems where conservation priorities diverge from community diversity priorities [49]. Climate models indicate that freshwater habitats in arid regions will be increasingly stressed by changes in temperature and precipitation, with detrimental consequences for quantity and quality of habitats [33]. Endemic species on the Arabian Peninsula face threats from climate change which can affect habitat quality, reduce ranges, and influence the effectiveness of management and conservation policies to safeguard endemic species and their ecosystems [30]. To help close knowledge gaps will require integrated approaches that combine predictive species distribution modelling with assessing habitats in the field and monitoring biodiversity [50]. These types of frameworks reveal complex ecological relationships between environmental variables, species occurrence patterns, and community-level biodiversity metrics to inform more sophisticated conservation decisions [51]. The design of habitat indices (HSI) that consider more than one environmental variable in a study system can lead to more mechanistic understanding of species-environment relationships, while providing quantitated assessments of habitat quality [52–54]. To facilitate effective conservation planning it is important to evaluate the representativeness of current protected area networks for freshwater habitats and biodiversity. Oman has a complete protected area system with 31 reserves designated by 2024 (informed with Environment Authority of Oman), and represents ecosystems from coastal wetlands to mountain reserves. However, we do not know how effective the network is at protecting freshwater fish habitat and endemic species, especially because aquatic species have unique spatial considerations due to the need for connectivity between habitat patches and the natural flow regimes found within wadi systems.

This research fills these crucial knowledge gaps by applying an integrated approach that implements predictive species distribution modeling [55], detailed field-based habitat suitability assessment, and a systematic evaluation of protected area effectiveness [56] to inform the present and future conservation needs of endemic killifish in Oman. Our research follows a comprehensive systematic workflow and has three interrelated aims (Fig 1): First, we use MaxEnt modeling to predict current and future habitat suitability for endemic/native *Aphaniops* species (*Aphaniops kruppi* and *Aphaniops stoliczkanus*) under three shared socioeconomic pathways SSP1–2.6, SSP3–7.0, and SSP5–8.5 for the periods 2011–2040, 2041–2070, and 2071–2100, and we will identify range shifts, habitat contraction, and conservation priority areas across differing emission pathways. Second, we develop and validate habitat suitability indices for *Aphaniops* spp. at 12 field study sites in the Hajar Mountains and examine the relationship between habitat quality and patterns of aquatic biodiversity to better understand how site-specific habitat optimization contributes to a greater ecosystem conservation framework. Third, we analyze the spatial dimensions of capacity for effectively protecting freshwater fish habitat by overlapping the density patterns of streams across Oman with the boundaries of protected areas and assessing how well the freshwater habitat is incorporated into the present conservation network and priorities for enhancing conservation of priority streams. The conservation goals of this study respond to urgent conservation needs, given the growing pace of environmental change and the extinction risk faced by specialists in fragmented freshwater systems, like the isolated arid environment of Oman [3,57]. Our analyses add to the growing literature on systematic conservation planning [56] and

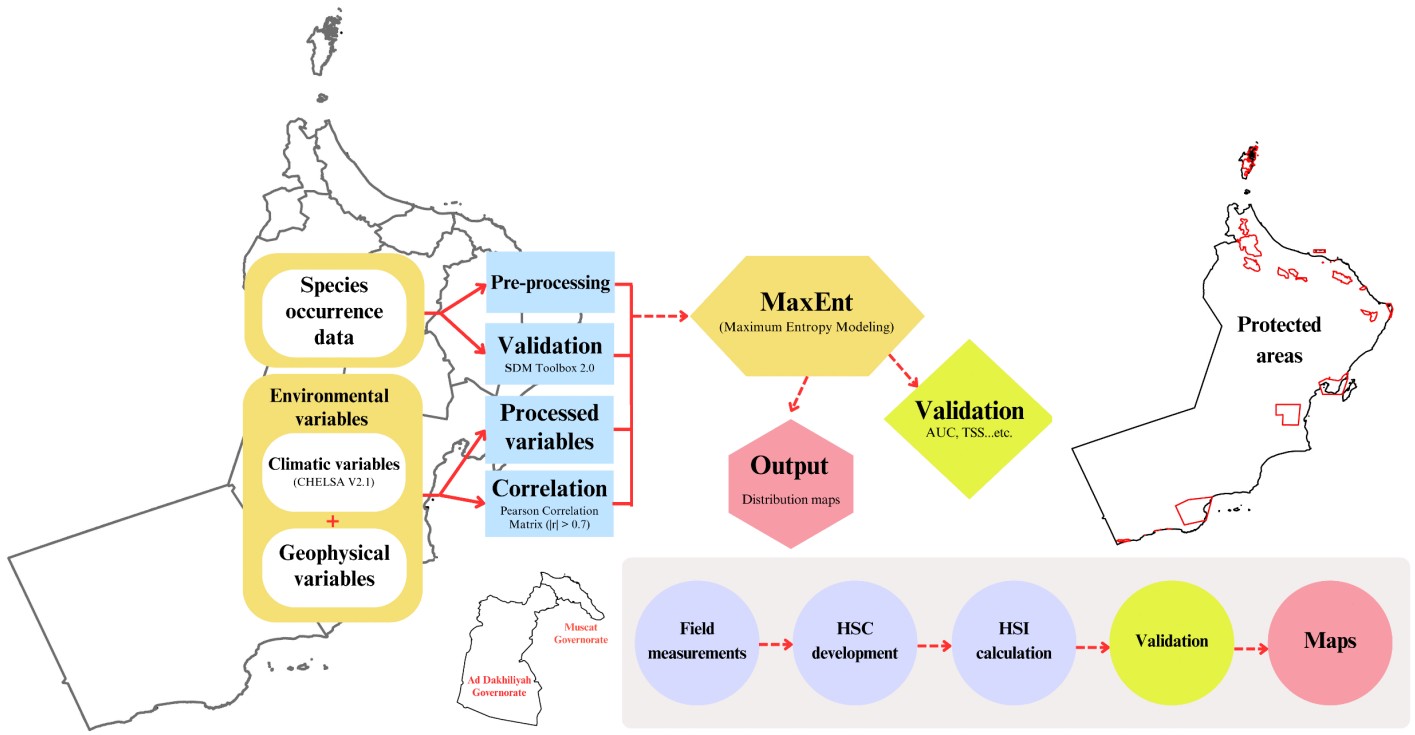

**Fig 1. Systematic methodology framework.**

provide evidence-based recommendations for the protection of freshwater biodiversity in arid environments. The results of this study can help guide informed decisions for conservation management in the region.

## 2. Methodology

This study follows a comprehensive systematic workflow, illustrated in Fig 1, which combines three main components. First, we use MaxEnt modeling to determine projected present and future potential habitat suitability for *A. kruppi* and *A. stoliczkanus* in response to the three future climate scenarios (SSP1–2.6, SSP3–7.0, SSP5–8.5) and the three time periods (2011–2040, 2041–2070, 2071–2100), and identify where these species are likely to shift their ranges and what areas would be best to protect. Second, we develop and validate a habitat suitability index using systematic field assessments conducted at 12 sites within the Hajar Mountain range, assessing the relationship between habitat quality and aquatic species diversity. Third, we assess the conservation status of freshwater habitats in Oman's protected areas using a spatial gap analysis by overlaying stream density with existing protected area boundaries.

### 2.1. Species distribution modeling of *A. kruppi* and *A. stoliczkanus*

  **2.1.1. Study area.** We examined the distribution of two species of killifish, native *Aphaniops stoliczkanus* (Day 1872) and the endemic *Aphaniops kruppi* (Freyhof, Weissenbacher & Geiger 2017) [10,11] throughout Oman's wadi systems spanning from southern Dhofar to northern and eastern drainages [10]. Their occurrence spans two distinct eco-regions: The Oman Mountain eco-region with the designation ID 443 stretches over the southeastern Arabian Peninsula including Oman and United Arab Emirates (UAE) and is surrounded by various water bodies and the Rub' al Khali Desert while the

Southwestern Arabian Coast eco-region designated as ID 439 reaches into Oman at the southern edge of the peninsula [58,59]. The distribution pattern demonstrates Oman's ephemeral and perennial waterways ecological connectivity which preserves aquatic biodiversity in dry regions [37].

Oman experiences regional differences in temperature patterns. The temperature range in Northern Oman varies between 32–48°C during May to September and falls to 26–36°C from October through April. During summer months coastal areas experience high temperatures up to 46°C along with humidity levels above 90%. The interior plains experience severe climate conditions as summer temperatures reach above 50°C followed by winter temperatures which remain below 23°C and above 15°C. The Dhofar region maintains a consistent temperature range of 30–35°C throughout the year while both Dhofar and the highland region experience moderate temperatures year-round [23].

Mountain ranges cover about 15% of Oman's entire land area and significantly impact both its geographical features and hydrological systems where they stand out as the primary source of fresh renewable water [23]. The Al Hajar Mountains stretch over a 700 km arc from Musandam in the north to Ras Al Hadd in the east in northern Oman [21]. The Jabal Shams stands as its highest summit at 3,075 m above sea level. Whereas, the Dhofar Mountains located in southwestern Oman contain peaks that rise between 1,000 and 2,000 meters above mean sea level [10,23]. The Hajar mountain range contains numerous spring-fed streams called wadis even though it receives very little and irregular rainfall [60]. The rainwater gathers in the mountain's subterranean aquifers and emerges occasionally as small lakes or creeks [61]. The region sustains multiple fish species within its freshwater environments which consist of wadis, pools, streams, falaj systems, and springs [12].

S1 Fig presents three spatial maps of Oman including (A) a digital elevation model that displays topographical variations throughout the landscape along with (B) the distribution of average annual temperature and (C) patterns of annual precipitation throughout the country.

**2.1.2. Species occurrence data.** We collected occurrence records for *Aphaniops kruppi* (40 points) and *Aphaniops stoliczkanus* (90 points) from multiple sources for this study. Among these were field surveys conducted earlier in the Oman Mountain ecoregion (ID 443) and the Southwestern Arabian Coast ecoregion (ID 439) [59], publicly accessible datasets from the Global Biodiversity Information Facility (GBIF: https://www.gbif.org/; [62–66]), and recent publications [10–12,27,30,67]. Fig 2 and S1 Table, illustrates the distribution of these records.

Prior to analysis, the data was cleaned to ensure accuracy by removing duplicate entries and filtering out geographic outliers outside the study area in ArcGIS [68,69]. To reduce spatial autocorrelation (clustering bias) and balance data quality with sample size, we applied 1 km spatial filtering, retaining only one record per 1 km according to earlier ecological research [37,42,70–72]. This filtering removed sampling biases potentially caused by over-sampling of more accessible areas (i.e., wadis close to roads or settlements) while maintaining sufficient sample sizes for model training (*A. kruppi*: n = 40; *A. stoliczkanus*: n = 90). The 1 km filtering distance was chosen to match the spatial resolution of our environmental predictors (~1 km), thereby ensuring that occurrence records and environmental data are spatially aligned [72].

Due to the low number of occurrences of *A. kruppi*, we decided against using spatial blocking for model evaluation to ensure sufficient data for cross-validation [73,74]. Therefore, we utilized MaxEnt, as it has been demonstrated to be capable of modeling species with limited occurrence data [75,76].

We determined both species' accessible area (M) by the FEOW eco-region [59] for the ID 443 and 439, which represent the geographical area that is accessible to the dispersal limited native/endemic species within fragmented wadi systems. This method was proposed as a way to define biologically relevant calibration areas with regard to dispersal limitations and biological/physical barriers [77].

**2.1.3. Environmental data.** The distribution of freshwater fish is affected by a variety of environmental factors operating at different spatial scales [38,41]. We selected predictors related to climate, topography, and hydrology at a (30 arcsec~1 km) resolution [40] to model suitable habitats for *A. kruppi* and *A. stoliczkanus* (Table 1).The bioclimatic data (1979–2013) were sourced from CHELSA V2.1 (https://chelsa-climate.org), a high-resolution (~1 km) global climate

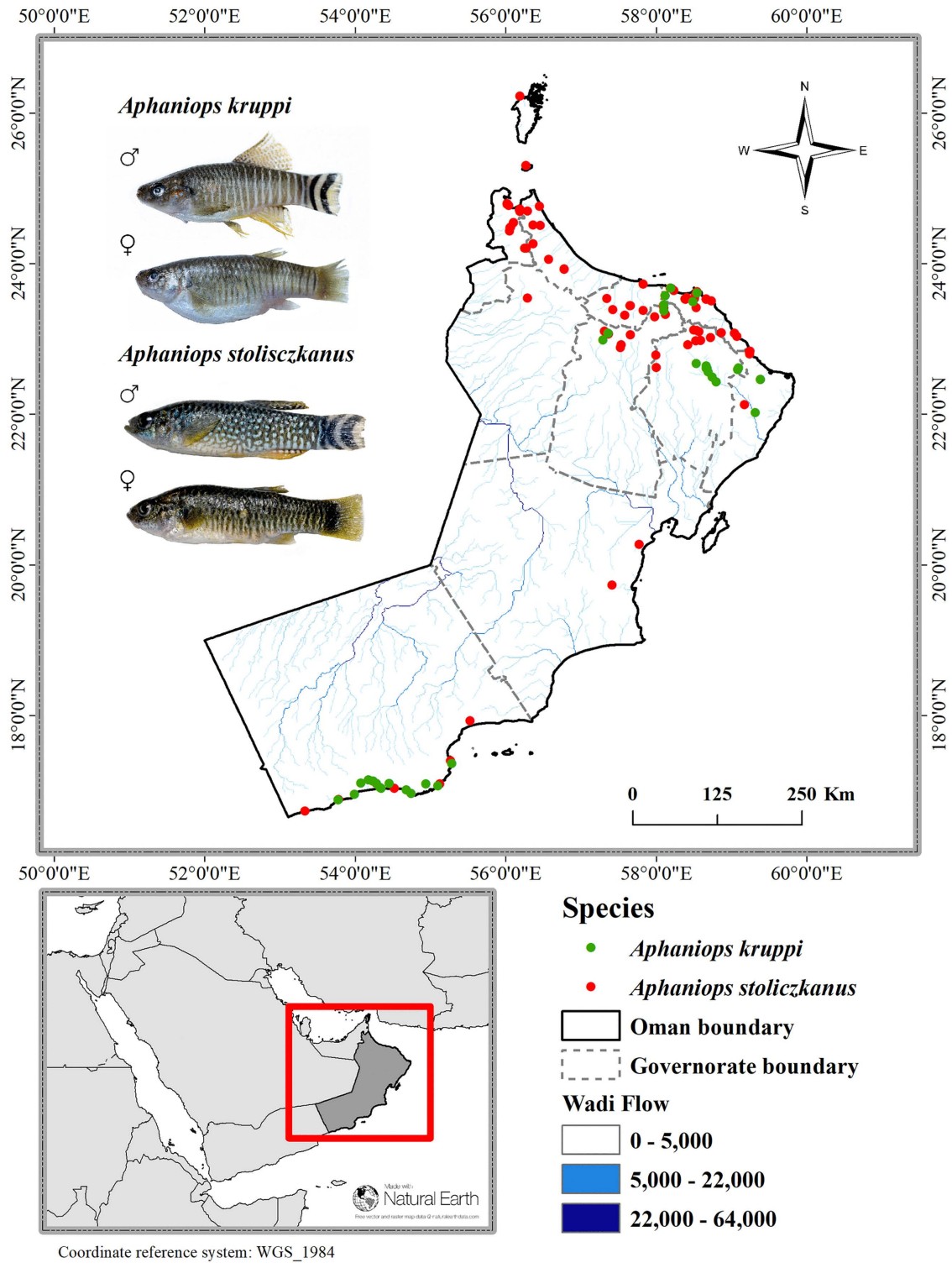

**Fig 2. Distribution of *Aphaniops kruppi* and *Aphaniops stoliczkanus* in Oman: species ranges and wadi flow capacity.**

**Table 1. List of predictor variables selected for the *Aphaniops* species distribution model.**

| Categories | Sub-categories | Variable | *Aphaniops kruppi* | *Aphaniops stoliczkanus* | Unit | Source |
|---|---|---|---|---|---|---|
| Climatic variables | Bio-climatic variables | Bio2 | * | * | °C | CHELSA-Climatologies at high resolution for the earth's land surface areas-V2.1 [89,90] |
| | | Bio8 | * | * | °C | |
| | | Bio9 | | * | °C | |
| | | Bio10 | | * | °C | |
| | | Bio11 | * | * | °C | |
| | | Bio12 | * | | kg m$^{-2}$ month$^{-1}$ | |
| | | Bio13 | | * | kg m$^{-2}$ month$^{-1}$ | |
| | | Bio14 | * | | kg m$^{-2}$ month$^{-1}$ | |
| | | Bio15 | * | * | kg m$^{-2}$ | |
| | | Bio17 | | * | kg m$^{-2}$ month$^{-1}$ | |
| | | Bio18 | | * | kg m$^{-2}$ month$^{-1}$ | |
| | | Bio19 | * | * | kg m$^{-2}$ month$^{-1}$ | |
| | | Cmi_m | * | * | kg m$^{-2}$ month$^{-1}$ | |
| | | Cmi_r | * | * | kg m$^{-2}$ month$^{-1}$ | |
| | | Hurs_m | * | * | % | |
| | | Hurs_r | * | * | % | |
| Geophysical variables | Topography | DEM | * | * | m | WorldClim 2.1 elevation User creation |
| | | SL | * | * | m | |
| | Soil | Soil | * | * | NA | Harmonized World Soil Database v 2.0 [88] |
| | Hydrology | FEOW | * | * | NA | WWF-FEOW database [59] |
| | | FA | * | * | NA | ArcGIS (10.8.2) |
| | | TWI | * | * | NA | ArcGIS (10.8.2) |
| | | STI | * | * | NA | ArcGIS (10.8.2) |
| | | SPI | * | * | NA | ArcGIS (10.8.2) |
| | | TRI | * | * | NA | ArcGIS (10.8.2) |

Bio2 = Mean diurnal air temperature range; Bio8 = Mean daily mean air temperatures of the wettest quarter; Bio9 = Mean daily mean air temperatures of the driest quarter; Bio10 = Mean daily mean air temperatures of the warmest quarter; Bio11 = Mean daily mean air temperatures of the coldest quarter, Bio12 = Annual Precipitation amount; Bio13 = Precipitation amount of the wettest month; Bio14 = Precipitation amount of the driest month; Bio15 = Precipitation seasonality; Bio17 = Mean monthly precipitation amount of the driest quarter; Bio18 = Mean monthly precipitation amount of the warmest quarter; Bio19 = Mean monthly precipitation amount of the coldest quarter; Cmi_m = Mean monthly climate moisture index; Cmi_r = Annual range of monthly climate moisture index; Hurs_m = Mean monthly near-surface relative humidity; Hurs_r = Annual range of monthly near-surface relative humidity; DEM = Digital Elevation Model; SL = slope; Soil = soil layer; FEOW = Freshwater eco-regions of the world; FA = Flow accumulation; TWI = Topographical wetness index; STI = Sediment transport index; SPI = Stream power index; TRI = Terrain ruggedness index.

* Selected variables used to model each species.

dataset. To project future habitat suitability, we used the GFDL-ESM4 (Geophysical Fluid Dynamics Laboratory Earth System Model version 4) Global Climate Model from CMIP6 [78–80] under three shared socioeconomic pathways SSP1–2.6, SSP3–7.0, and SSP5–8.5 [81] across four time periods: 1981–2010, 2011–2040, 2041–2070, and 2071–2100. While multi-model ensemble methods are generally preferred as they provide a better means of accounting for the inter-model variability of different models; employing a single, well-developed general circulation model (GCM), such as GFDL-ESM4, is methodologically sound and provides internally consistent projections for assessing climate change impacts on species distributions [82,83]. Additionally, GFDL-ESM4 demonstrated skill in simulating climate dynamics for the Arabian Peninsula [32]. In Section 2.1.6, we describe a comprehensive three-part uncertainty framework for quantifying and managing the uncertainty associated with using a single GCM.

To minimize collinearity, we applied a Pearson Correlation Matrix (threshold: |r| > 0.7) to 19 bioclimatic variables, and the climate moisture index (cmi) – which is derived by subtracting precipitation amounts from potential evapotranspiration–, and the near-surface relative humidity (hurs) [79], thereby removing highly correlated predictors [31,37,50]. We selected pairwise correlation as opposed to Variance Inflation Factor (VIF) for a number of methodological reasons; (1) VIF is typically only applicable to presence-absence data structures and we are working with a presence background framework using MaxEnt, so VIF would be an inappropriate way to calculate variable redundancy in this case [84]; (2) Pairwise correlation has been shown to represent the common practice for selecting variables for inclusion in MaxEnt models prior to model building [85]; and (3), MaxEnt uses a variety of regularization parameters that can help reduce multicollinearity inherent in many MaxEnt models by penalizing very complex combinations of features [86]. This process yielded 11 bioclimatic variables for *A. kruppi* and 14 bioclimatic variables for *A. stoliczkanus* (Table 1).

We characterized habitats using nine geophysical variables (Table 1), including a digital elevation model (DEM) obtained from WorldClim 2.1 at 30 second spatial resolution (~1 km) and converted to ascii format so that it would be compatible with MaxEnt, along with derived metrics: slope (SL), topographic wetness index (TWI), flow accumulation (FA), sediment transport index (STI), stream power index (SPI), and terrain ruggedness index (TRI) [31,41], all of which were generated from DEM and processed in ArcGIS 10.8.2 Spatial Analyst hydrology tools [87]. Eco-region data from the WWF-FEOW database [59] helped identify ecological hotspots and assess climate change impacts. Soil properties were obtained from the Harmonized World Soil Database version 2.0 (HWSD) [88], a globally recognized resource for soil and land management studies.

**2.1.4. Species distribution modeling with MaxEnt.** We used Maximum Entropy (MaxEnt v3.4.4) [91], a machine-learning algorithm based on the maximum entropy principle, to model current and future suitable habitats for *Aphaniops kruppi* and *A. stoliczkanus*. The model predicts species distributions by combining presence records with background environmental variables, constrained to the most probable geographic extent [76]. Moreover, both continuous and categorical variables and their exchanges can be used within this model. In order to prevent over-fitting, Maxent algorithm limits model complexity through regularization [55,92]. Additionally, MaxEnt is capable of creating consistent models even with a small sample size [37,41,76,84,93]. We used MaxEnt default settings, to generate 10,000 random background data points that are spatially limited to the entire defined accessible area (M) where a species can potentially be present and utilize environmental resources available at each location. This is done to prevent biased habitat suitability predictions due to sampling historical un-accessible geographic locations to the species [77,94]. A 10,000-point background sample was sufficient for describing all potential environmental characteristics on M, while remaining computationally efficient [95].

We employed MaxEnt using the default regularization ($\beta = 1$), 15 bootstrapped replications of the data, and an auto-feature class to provide optimal performance and avoid over-fitting; we also utilized a logistic function to produce the suitability values (0–1) for each cell. The use of bootstrapping was especially useful for our low sample size (i.e., *A. kruppi*, n = 40) since it provided a measure of the robustness of the uncertainty associated with the estimation of the distribution of the species [55,76,95–97]; whereas the use of regularization and the application of automatic feature selection were utilized to control the complexity of the model [84,92,98]. We ran the model with 500 iterations per replicate, splitting the occurrence data into training (70%) and testing (30%) subsets using a random test percentage of 30% and a training presence threshold of 10th percentile. Based on established standards for species distribution modeling, we evaluated model performance on independent test data not used during model training [73,99].

Then, we projected habitat suitability under three future periods (2011–2040, 2041–2070, 2071–2100) and compared these to the baseline (1981–2010). Suitability was classified into four categories: absent (0.0–0.25), low suitability (0.25–0.50), medium suitability (0.50–0.75), and high suitability (0.75–1.00). The use of this equal interval approach was made because of its simplicity and interpretability when comparing continuous suitability data for the purposes of species distribution modeling, which has been used in a variety of plant [100], mammal [101], and marine fish [102] studies. Other classifications are possible and have been proposed, such as classification intervals defined by statistical measures of

model performance [103]; however, equal interval classification provides consistency and comparability among studies. It also is important to recognize that sites near classification boundaries (for example 0.24–0.26 or 0.49–0.51) should be viewed cautiously, as these areas of classification ambiguity represent zones where relatively minor differences in either environmental characteristics or model parameters can cause a shift in the classification category assigned. Sites located in these areas may require targeted field validation to provide assurance for suitability predictions.

Spatial changes in suitable areas were quantified using ArcGIS 10.8.2. Hence, we created 10 suitability maps featuring the current and future scenarios, and periods for each species.

**2.1.5. Model validation.** Model validation was assessed using the Area Under the Curve (AUC) of the Receiver Operating Characteristic (ROC) quantifies overall model discrimination ability, ranging from 0 to 1, where 0.5 indicates random prediction and values approaching 1 denote high accuracy [43,104–106]. The True Skill Statistic (TSS) evaluates the balance between sensitivity and specificity, ranges from −1 to +1, the TSS value of +1 indicates perfect agreement, 0 indicates no better than random predictions, and −1 indicates perfect disagreement with scores ≥0.5 indicating robust performance [107,108].

$$Sensitivity = \frac{TP}{TP + FN} \tag{1}$$

$$Specificity = \frac{TN}{TN + FP} \tag{2}$$

$$TSS = Sensitivity + Specifity - 1 \tag{3}$$

For threshold-dependent validation, we employed three methods the default threshold (0.5), Maximum Training Sensitivity plus Specificity (MaxSS), and Minimum ROC Distance (MinROCdist), which identifies the threshold closest to the perfect classification point on the ROC curve. For each method, we calculated:

$$Accuracy = \frac{TP + TN}{TP + TN + FP + FN} \tag{4}$$

$$Balanced\ Accuracy = \frac{Sensitivity + Specifity}{2} \tag{5}$$

$$Predicted\ Prevalence = \frac{TP + FP}{N} \tag{6}$$

$$Observed\ Prevalence = \frac{TP + FN}{N} \tag{7}$$

where TP is *true positives*, FP is *false positives*, TN is *true negatives*, FN is *false negatives*, and N is the sample size of the test set [108,109].

In addition to the threshold dependent metrics mentioned above, we examined the model's calibration using an assessment approach consistent with the Continuous Boyce Index (CBI) to evaluate models' calibration. CBI is based on a correlation between habitat suitability values assigned by a model for each location along a suitability gradient and frequency of occurrence [110–112]. However, we developed a robust version of this assessment, comparing the mean predicted

habitat suitability at occurrence locations versus background locations. This method calculates a normalized performance ratio, defined as (mean presence/mean background − 1)/ (mean presence/mean background + 1); this ratio values ranging from −1 to +1. Similar to standard CBI interpretation, ratios greater than zero reflect that modeled habitat suitability values are well correlated with observed species occurrences; higher positive ratios indicate better calibration of the model [110–112]. Our approach retains the ecologically meaningful aspects of standard CBI analysis while being more robust to small sample sizes and specific data distributions in our study. In order to ensure statistical robustness, we reported means and standard deviations across 15 replicates. For this analysis and prevalence consistency checks (flagging deviations >10% from observed prevalence) we used the R packages PresenceAbsence and ecospat.

**2.1.6. Uncertainty and extrapolation analysis.** To address uncertainty of projections, we implemented three methods to provide a comprehensive assessment of uncertainty in our model outputs. (1) To understand both the variability of our predictions and their reliability through time, we ran 15 bootstrapped model replicates to establish how well our predictions were stabilized. (2) We conducted a Multivariate Environmental Similarity Surface (MESS) analysis to determine areas where future climate projections represent novel conditions (extrapolation) compared to the current environmental space used for model training [113]. A negative MESS value indicates extrapolation risk, where model projections are less reliable [113,114]. The MESS analysis was completed for all 9 future scenarios using the dismo package within R [115]. We visualized MESS values using a continuous color gradient (red: −100 to blue: +100) and classified MESS values into three ecological categories based on study-specific thresholds to facilitate ecological interpretation: strong extrapolation (MESS < −10), moderate extrapolation (−10 ≤ MESS < 0), and analog conditions (MESS ≥ 0). (3) Comprehensive Model Evaluation: As detailed in Section 2.1.5, we used independent test data and multiple validation metrics (AUC, TSS, CBI) to ensure robust model performance.

## 2.2. Habitat suitability assessment (12 Hajar Mountain sites)

**2.2.1. Study area.** We selected twelve freshwater wadis in Oman's Hajar Mountain ecoregion, representing four areas: Darsait (D), Al Amirat (A), Al Khoud (K) in Muscat Governorate, and Ain Wadhah (AW) in Ad Dakhiliyah Governorate (Fig 3). We categorized these wadis by stream position (upstream, midstream, downstream) and assigned them station codes: Darsait (D1-D3), Al Amirat (A1-A3), Surur/Fanja/Al Khoud (K1-K3), and Ain Wadhah (AW1-AW3). All of these sites originate from the Hajar Mountain eco-region with permanent streams throughout the year. Additionally, all sites are accessible with presence of native freshwater fishes. Eight sites feature sandy substrates mixed with gravel (10–90%), while roads, dams, and recreation areas constitute the most frequent human disturbances. Riparian vegetation density varies across sites, with midstream areas typically supporting the greatest coverage (S2 Table).

Recent research reported that *A. stoliczkanus* and *A. kruppi* mitochondrial haplotypes exist together in Wadi Fanja and Wadi Surur. These findings indicate substantial genetic linkages and possible introgressive hybridization between *Aphaniops* populations/species across Northern Oman [26,27]. Hence, *Aphaniops* spp. is considered the appropriate taxonomic designation for this habitat suitability assessment.

**2.2.2. Habitat suitability analysis for *Aphaniops* spp.** We developed a Habitat Suitability Index (HSI) model for *Aphaniops* spp. following Brooks' (1997) four step framework [52,116,117], collecting environmental and hydrological data on a quarterly basis from March 2022 to March 2023 across twelve wadis (A1-A3, AW1-AW3, D1-D3, K1-K3). Our initial step involved determining crucial habitat elements through the measurement of twelve environmental parameters across all locations. These included physical characteristics (depth ranging from 25−84 cm, width from 2.5–11.5 m, and velocity from 0.01–0.4 m/s), water chemistry (temperature between 28–31.6 °C, pH values of 7.9–8.6, electrical conductivity measuring 696.2–2609.5 µS/cm, total dissolved solids of 351.7–1331.2 mg/L, salinity between 0.33–1.33 ppt, dissolved oxygen levels from 5.8–10.4 mg/L, and biochemical oxygen demand ranging from 2.84–4.36 mg/L), water quality (turbidity between 1.1−22 NTU), and substrate quality (soil texture). We analyzed soil texture for particle size distribution using the Cosby pedotransfer function [118], then normalized the results to a 0–1 scale via logarithmic normalization [119] to align

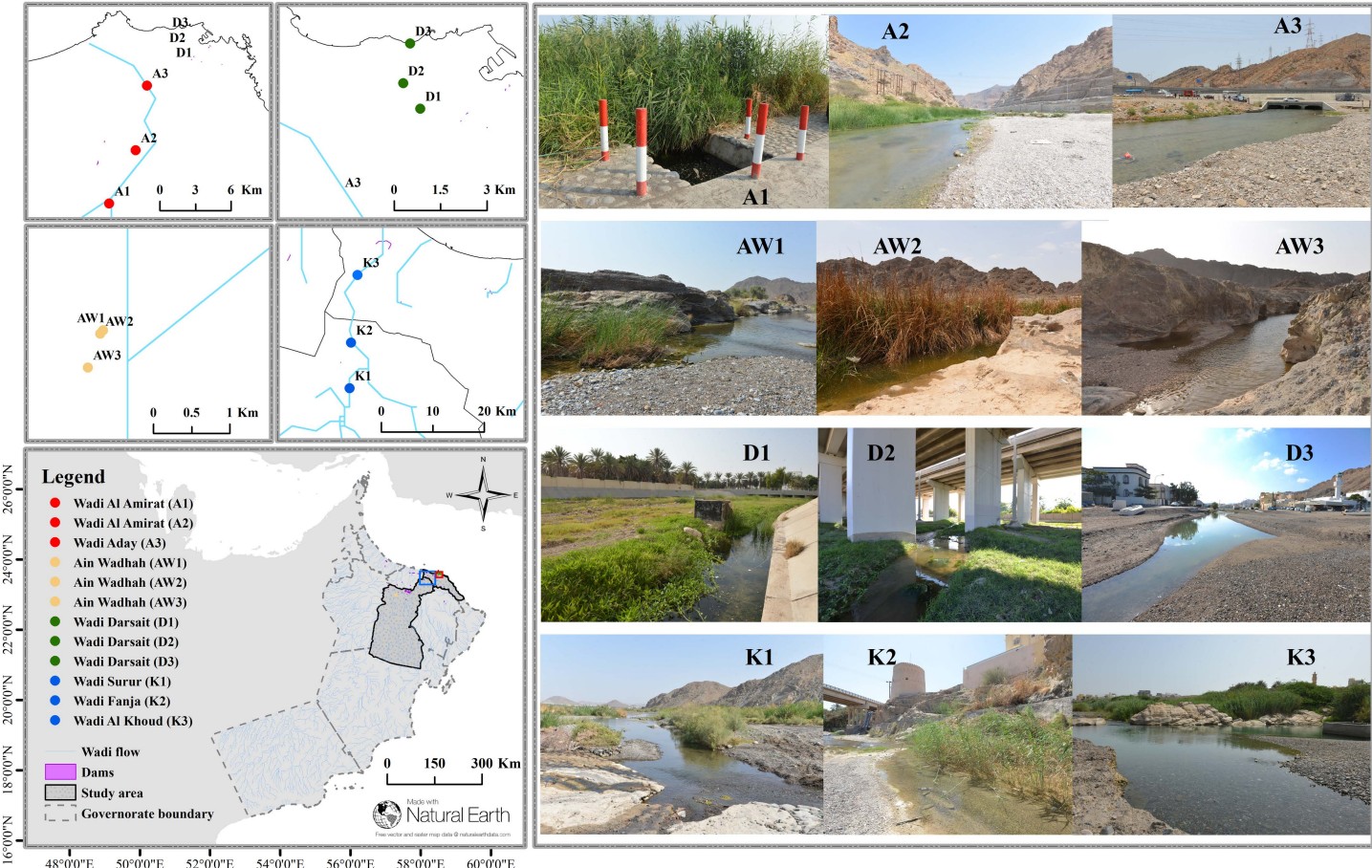

**Fig 3. Sampling locations and visual documentation of wadis in Northern Oman's Hajar Mountain ecoregion.**

with our Gaussian modeling framework. The *Aphaniops* population abundance assessment involved temporary capture methods using foldable fishing traps (mesh size 3 × 3 mm, consisting of 4 nets). Fish welfare was considered through the process of sampling fish. Once fish were collected, they were placed in holding tanks supplied with oxygen and the fish were counted and identified. After that, fish were released back to the location from which they were captured in a healthy condition. No fish were kept, sacrificed, subjected to experimental manipulation, or even killed through a brief observation process for identification. The number of *Aphaniops* fish demonstrated considerable variation (979−17,708 individuals) across different wadis, with Wadi Aday (A3) sustaining the largest populations and Wadi Darsait (D3) sustaining the smallest (S3 Table).

Second, we developed habitat suitability curves for each parameter using Gaussian functions, calculating weighted means ($\mu$) and standard deviations ($\sigma$) based on *Aphaniops* spp. abundance. Habitat suitability was modeled using the equation:

$$S(x) = \exp\left(-\frac{(x - \mu)^2}{2\sigma^2}\right)$$

(8)

where S(x) represents the suitability index (0−1) for parameter value x. All parameters were constrained to ecologically plausible ranges to avoid extrapolation beyond observed conditions. This produced suitability scores between 0−1 for

each parameter [120]. We then calculated the relative importance of each environmental parameter by comparing mean suitability values and their variability (standard deviation) across all sites.

Third, we utilized the Geometric Mean habitat suitability index instead of an arithmetic mean for parameter combination to obtain more conservative Habitat Suitability Index values since it reduces the overall score to zero whenever one parameter proves unsuitable thereby penalizing habitats with any critical limitations [53,121,122]. We also conducted a comparative assessment of HSI values with and without substrate texture to evaluate the specific contribution of this parameter to overall habitat suitability [123].

Fourth, we validated our model through statistical testing (Shapiro-Wilk), biological relevance assessment (comparing HSI with biodiversity metrics), and independent validation (AUC and TSS). For comprehensive model evaluation, we calculated the Boyce Index and generated a confusion matrix to assess classification performance, including sensitivity, specificity, precision, recall and F1 score metrics [104]. The Boyce Index is a useful presence-only, threshold-independent assessment metric that avoids pseudo-absence data assumptions [111].

Additionally, we used Non-metric Multidimensional Scaling to visualize community composition differences between sites. Through correlation analyses we examined the relationship between HSI values and various biodiversity measures including Shannon diversity, Simpson's diversity, and evenness to determine our model's biological significance. Finally, we applied the quantile classification method following Edosa and Erena [124] to classify HSI values into five zones based on data distribution: The quantile classification method divided HSI values into five zones: unsuitable for the bottom 20%, less suitable for 20–40th percentile values, moderately suitable for 40–60th percentile values, suitable for 60–80th percentile values, and highly suitable for the top 20% [124]. This data-driven approach ensures objective habitat quality assessment based on the actual HSI distribution within our study area.

To address potential over-fitting, and quantify uncertainty, we utilized bootstrap resampling (1000 iterations), to create 95% confidence intervals for correlation coefficients and Leave-One-Out Cross-Validation (LOOCV), as a means of evaluating model predictive performance for use with smaller ecological data sets [73,111,125–127].

We processed our data with R (version 4.4.3) packages dplyr, tidyr and magrittr; applied EnvStats and nortest for statistical validation and boot for bootstrap analysis; and used ggplot2, corrplot scales, extrafont, and ggpubr for visualization. ROCR, PresenceAbsence, ecospat, caret, and Metrics were used to develop ecological models and validation metrics. All packages were managed using pacman, and writexl facilitated data export.

## 2.3. Conservation prioritization

Using protected area boundaries obtained from the Geographic Information System (GIS) Department, Environment Authority, Sultanate of Oman (31 designated reserves by 2024), we conducted a spatial analysis to evaluate freshwater fish habitat protection in Oman. In order to derive wadi flow networks, we processed Digital Elevation Model (DEM) data using ArcGIS 10.8.2. Flow direction and accumulation were calculated using the Spatial Analyst toolbox. Using the Field Calculator function, we calculated stream density as total stream length divided by study area. As a final step, we analyzed how Oman's conservation network represents freshwater habitats by overlaying the stream density layer with protected area boundaries.

To establish the conservation criteria and quantifiable targets we performed two complementary spatial analyses:

**2.3.1. Individual protected area analysis.** We analyzed habitat representation in Oman's 31 protected areas through overlaying of species distribution models for *A. kruppi* and *A. stoliczkanus* on their respective protected areas. High-habitat suitability was determined using the 90th percentiles threshold. For each protected area, we calculated the area and percentage of suitable habitat for each species, the wadi density classification (high: > 0.0008; medium: 0.0004–0.0008; low: < 0.0004 km/km²), and assigned a conservation priority score. All analyses were completed in R version 4.4.3 utilizing spatial packages (raster, sf, terra, dplyr, and exactextractr) for accurate area-weighted calculations in WGS84 coordinate reference system.

**2.3.2. Quantifying conservation gaps and establishing targets.** The gap in protection was evaluated using Oman's National Environment Strategy (2021–2040), developed by the Environment Authority in alignment with Oman Vision 2040 [128] and the Kunming-Montreal Global Biodiversity Framework targets (CBD 30x30) [129,130]. The national target for developing protected areas in wetlands for the year 2040 is 7.5% of the total habitat (25% of the 30% goal of the IUCN 30%), measured by the indicator "Percentage of beneficiary protected areas (Ramsar sites) of total wetland area", with interim milestones of 10% by 2025, 15% by 2030, and 20% by 2035. Oman currently has 3 designated Ramsar sites (wetlands of international importance). We applied these national wetland protection targets to freshwater fish habitats by determining the area of protected high-suitability habitat for each species, calculating the rate of protection, and quantifying gaps from both national (7.5%) and international (30%) targets. Protection of wadis was evaluated based on specific density-based targets: 50% for high-density (critical connectivity corridors), 30% for medium-density (landscape connectivity), and 10% for low-density (minimal conservation priority). The output of the analyses was developed for conservation planning purposes using both ArcGIS (10.8.2) and R version 4.4.3 utilizing spatial packages (raster, sf, dplyr).

## 3. Results

### 3.1. Species distribution model performance

MaxEnt models of distribution for the *Aphaniops* killifish species demonstrated very good discrimination performance in an independent test set. Notably, this excellent discrimination performance was achieved with a relatively low sample size for the endemic *A. kruppi* (n = 40) and demonstrates MaxEnt's potential to identify significant relationships between species and environments using limited occurrence data [75,76]. For *A. kruppi*, both MaxSS and MinROCdist threshold selection methods resulted in identical values for the threshold (0.167 ± 0.131) and therefore identical performance metrics (S4 Table). This convergence phenomenon in species distribution modeling, occur in high-performing models (AUC > 0.9) when different optimality criteria may select identical thresholds due to the ROC curve's structure near the optimal operating point [103].

The AUC results revealed that *A. kruppi* outperformed *A. stoliczkanus* with values of 0.974 ± 0.014 versus 0.950 ± 0.018, while TSS scores showed 0.915 ± 0.090 compared to 0.817 ± 0.073, as shown in S2 Fig and S4 Table. The default threshold (0.5) achieved high specificity in *A. kruppi* with values of 0.993 ± 0.003 compared to *A. stoliczkanus* (0.990 ± 0.003) but showed low sensitivity (0.52 ± 0.126 vs. 0.448 ± 0.129). In contrast, MaxSS and MinROCdist thresholds led to significant sensitivity gains in *A. kruppi* versus *A. stoliczkanus* (0.973 ± 0.070 and 0.915 ± 0.069) and better-balanced accuracy (0.957 ± 0.045 and 0.909 ± 0.037), respectively, as demonstrated in S2 Fig and S4 Table.

The model for *A. stoliczkanus* showed more significant prevalence mismatches (73 versus 57 for *A. kruppi*), indicating more substantial calibration difficulties, which may stem from ecological niche breadth variations or environmental gradient data quality differences.

The CBI results showed that the models performed exceptionally well for both species: with values of 0.909 ± 0.025 for *A. kruppi* and 0.872 ± 0.028 for *A. stoliczkanus* (S4 and S5 Tables). These positive, high values support our earlier discrimination assessment (i.e., AUC and TSS values), which show there was an extremely high degree of agreement between the predicted suitability gradient for each species and their respective occurrence pattern. The better CBI value for *A. kruppi* was also in line with the higher AUC and TSS values for *A. kruppi*, thereby reinforcing the reliability of predictions for this narrowly-distributed species.

### 3.2. Variables contribution to the suitability of *Aphaniops* species

Variable contribution analysis demonstrated unique environmental factors shaping the range patterns of both *Aphaniops kruppi* and *Aphaniops stoliczkanus* (S3 Fig). The top environmental factors impacting *A. stoliczkanus* are Bio19 (mean monthly precipitation amount of the coldest quarter), STI (sediment transport index), SPI (stream power index), and Slope at 31%, 20.2%, 13.3%, and 10.5%, respectively. On the other hand, *A. kruppi* primarily exhibited strong connections

to Cmi_m (mean monthly climate moisture index) and Bio2 (mean diurnal air temperature range) at 39.9% and 18.3%, respectively. Followed by STI at 8.4% and TWI (topographical wetness index) at 7.4%.

### 3.3. Predicted current and future responses to suitability

Table 2 display *Aphaniops kruppi*'s and *Aphaniops stoliczkanus*' current and projected habitat suitability for SSP1–2.6, SSP3–7.0 and SSP5–8.5 climate scenarios. Under three separate climate scenarios the potential suitable habitats for both species extend over multiple time intervals including 1981–2010 as baseline/current followed by 2011–2040, 2041–2070 and 2071–2100. The predicted areas are measured in square kilometers (km²) and classified into four suitability categories: The suitability categories Absent (AB), Low-suitability (LS), Medium-suitability (MS), and High-suitability (HS) derive from a suitability score that span from 0 to 1.

The total area of different habitat suitability classifications for *A. kruppi* demonstrates distinct patterns when compared with its current distribution. The LS classification under SSP1–2.6 shows a rising trend growing from 0.36% during 2011–2040 to 0.55% for the 2041–2070 period and reaching 0.65% by 2071–2100. The MS classification shows a steady increase from 0.10% in 2011–2040 to 0.14% in 2041–2070 before reaching 0.16% in 2071–2100. The HS classification maintains its level at 0.04% during initial periods then shows a slight reduction to 0.03% in 2071–2100. Under SSP3–7.0, LS values demonstrate substantial growth from 0.32% in 2011–2040 to 0.74% in 2041–2070 then increasing to 1.07% in 2071–2100. The MS classification increases from 0.08% in 2011–2040 to 0.18% in 2041–2070, and further to 0.20% in

**Table 2. Future habitat suitability for *Aphaniops* species under climate change scenarios. Projections for *A. kruppi* and *A. stoliczkanus* across multiple emission scenarios.**

| Period | Climate Scenario | Species | Predicted area (km²) | | | | Rate change from baseline (%) | | | |
|---|---|---|---|---|---|---|---|---|---|---|
| | | | AB | LS | MS | HS | AB | LS | MS | HS |
| 1981-2010 | Baseline | *A. kruppi* | 4,168,706.4 | 69,619.8 | 18,730.2 | 5,933.4 | NA | NA | NA | NA |
| | | *A. stoliczkanus* | 4,120,979 | 103,137 | 27,602.4 | 5,561.4 | NA | NA | NA | NA |
| 2011-2040 | SSP1–2.6 | *A. kruppi* | 4,201,814.4 | 86,006.4 | 23,343 | 7,626 | −0.50 | 0.36 | 0.10 | 0.04 |
| | | *A. stoliczkanus* | 4,124,587 | 102,337.2 | 25,761 | 4,594.2 | 0.08 | −0.02 | −0.04 | −0.02 |
| | SSP3–7.0 | *A. kruppi* | 4,204,734.6 | 84,183.6 | 22,468.8 | 7,402.8 | −0.43 | 0.32 | 0.08 | 0.03 |
| | | *A. stoliczkanus* | 4,123,006 | 103,936.8 | 25,947 | 4,389.6 | 0.05 | 0.02 | −0.04 | −0.03 |
| | SSP5–8.5 | *A. kruppi* | 4,198,782.6 | 88,461.6 | 23,789.4 | 7,756.2 | −0.57 | 0.42 | 0.11 | 0.04 |
| | | *A. stoliczkanus* | 4,110,247 | 113,106.6 | 28,867.2 | 5,059.2 | −0.25 | 0.23 | 0.03 | −0.01 |
| 2041-2070 | SSP1–2.6 | *A. kruppi* | 4,191,882 | 94,078.80 | 25,017 | 7,812 | −0.73 | 0.55 | 0.14 | 0.04 |
| | | *A. stoliczkanus* | 4,100,705 | 120,490.8 | 30,485.4 | 5,598.6 | −0.48 | 0.41 | 0.07 | 0.00 |
| | SSP3–7.0 | *A. kruppi* | 4,183,884 | 102,653.4 | 26,839.8 | 5,412.6 | −0.91 | 0.74 | 0.18 | −0.01 |
| | | *A. stoliczkanus* | 4,066,462 | 147,553.8 | 36,809.4 | 6,454.2 | −1.28 | 1.04 | 0.22 | 0.02 |
| | SSP5–8.5 | *A. kruppi* | 4,180,591.8 | 103,471.8 | 26,728.2 | 7,998 | −0.99 | 0.76 | 0.18 | 0.05 |
| | | *A. stoliczkanus* | 4,098,194 | 124,768.8 | 30,001.8 | 4,315.2 | −0.54 | 0.51 | 0.06 | −0.03 |
| 2071-2100 | SSP1–2.6 | *A. kruppi* | 4,186,822.8 | 98,728.8 | 25,798.2 | 7,440 | −0.84 | 0.65 | 0.16 | 0.03 |
| | | *A. stoliczkanus* | 4,071,949 | 144,001.2 | 35,321.4 | 6,007.8 | −1.15 | 0.96 | 0.18 | 0.01 |
| | SSP3–7.0 | *A. kruppi* | 4,169,469 | 116,863.8 | 27,825.6 | 4,631.4 | −1.25 | 1.07 | 0.20 | −0.03 |
| | | *A. stoliczkanus* | 4,132,697 | 105,015.6 | 18,525.6 | 1,041.6 | 0.28 | 0.04 | −0.21 | −0.11 |
| | SSP5–8.5 | *A. kruppi* | 4,192,123.8 | 98,896.2 | 23,305.8 | 4,464 | −0.72 | 0.66 | 0.10 | −0.04 |
| | | *A. stoliczkanus* | 4,123,304 | 111,246.6 | 21,576 | 1,153.2 | 0.05 | 0.19 | −0.14 | −0.10 |

AB = Absent (0–0.25), LS = low-suitability (0.25–0.5), MS = Medium-suitability (0.5–0.75), HS = High-suitability (0.75–1), All percentage changes calculated relative to the 1981–2010 baseline period. SSP1–2.6 represents a low emissions scenario, SSP3–7.0 a high emissions scenario, and SSP5–8.5 a highest emissions scenario.

2071–2100. The HS classification becomes negative, decreasing to −0.01% in 2041–2070 and −0.03% in 2071–2100. The LS area experiences an increase from 0.42% during 2011–2040 to 0.76% in 2041–2070 then faces a decline to 0.66% in 2071–2100 under SSP5–8.5. The MS classification reaches 0.18% in 2041–2070 but declines to 0.10% in 2071–2100. The HS classification demonstrates earlier fluctuations between 0.04% and 0.05% but reduces to −0.04% in 2071–2100.

Habitat dynamics for *A. stoliczkanus* exhibit substantial variations throughout every suitability classification. Under SSP1–2.6, the LS classification shows an increase from −0.02% in 2011–2040 to 0.41% in 2041–2070 followed by a rise to 0.96% in the period 2071–2100. The MS classification demonstrates a negative trend of −0.04% in 2011–2040 before increasing to 0.07% in 2041–2070 and further to 0.18% during 2071–2100. The HS classification displays minor fluctuations within a range of −0.02% and 0.01%. The SSP3–7.0 scenario shows significant shifts in the LS category which grew from 0.02% in 2011–2040 to 1.04% in 2041–2070 before falling back to 0.04% in 2071–2100. The MS classification reaches 0.22% in 2041–2070 before dropping sharply to −0.21% in 2071–2100 as the HS classification falls to −0.11%. Moreover, the LS classification demonstrates an increase from 0.23% in 2011–2040 to 0.51% in 2041–2070 under SSP5–8.5 before dropping slightly to 0.19% in 2071–2100. MS classification rates rose by 0.03% in 2011–2040 followed by a 0.06% gain in 2041–2070 then fell by 0.14% in 2071–2100. The HS classification demonstrates stability in initial periods before experiencing minor declines and ultimately reaching −0.10% in 2071–2100.

The spatial distribution patterns between the *Aphaniops* species show clear differences as depicted in habitat suitability maps (Figs 4 and 5). The suitable habitat for *A. stoliczkanus* extends extensively across northern Oman's coasts and includes low-suitability green patches along the northeastern shoreline. *A. kruppi* maintains a restricted distribution pattern with its main suitable habitats found in small isolated patches along southern coastal regions where yellow spots (medium suitability) and small red dots (high suitability) are visible in the southernmost region. *A. stoliczkanus* exhibits wider spatial coverage in northern areas compared to *A. kruppi* which maintains viable habitats in limited southern coastal zones. All climate scenarios show that these spatial differences remain constant while habitat suitability for each species changes in scope and condition throughout the future periods.

### 3.4. Climate projection novelty assessment

The MESS analysis revealed a very high extrapolation risk for the two species across all climate scenarios, with a wide range of species-specific vulnerabilities (see Table 3, S4 and S5 Figs). Of note was the significantly greater extrapolation risk exhibited by *Aphaniops kruppi* compared to *A. stoliczkanus*. Specifically, 85.26–94.92% of the study area was classified as having strong extrapolation risk for *A. kruppi*, whereas 62.07–92.64% of the study area was classified as such for *A. stoliczkanus*. Both species showed increasing extrapolation under higher emission pathways and later time periods. Furthermore, analog conditions were severely limited, never exceeding 11.47% for *A. stoliczkanus* and 6.87% for *A. kruppi*, indicating that most future climate conditions fall outside the current environmental envelope of these species. The spatial patterns visualized in the MESS maps (S4 and S5 Figs), show clear and striking extrapolation gradients for each species. Extensive red zones (strong extrapolation risk) dominate future projections particularly, for *A. kruppi*, whereas limited blue areas (analog conditions) are indicative of possible climate refugia.

A complementary bootstrap analysis supported the high predictive uncertainty for large areas of the projected ranges for both species (S6 Fig and S6 Table), with over 93% of the model pixels for each species exhibiting a high coefficient of variation (CV > 0.5).

### 3.5. Assessment of habitat suitability for *Aphaniops* spp

**3.5.1. Model performance evaluation.** Assessment of habitat suitability for *Aphaniops* spp. exhibits strong performance metrics. The Boyce Index measurement of 0.894 demonstrates significant correlation between habitat suitability predictions and observed species occurrences (S7 Fig). With an AUC value of 0.778 the model shows acceptable

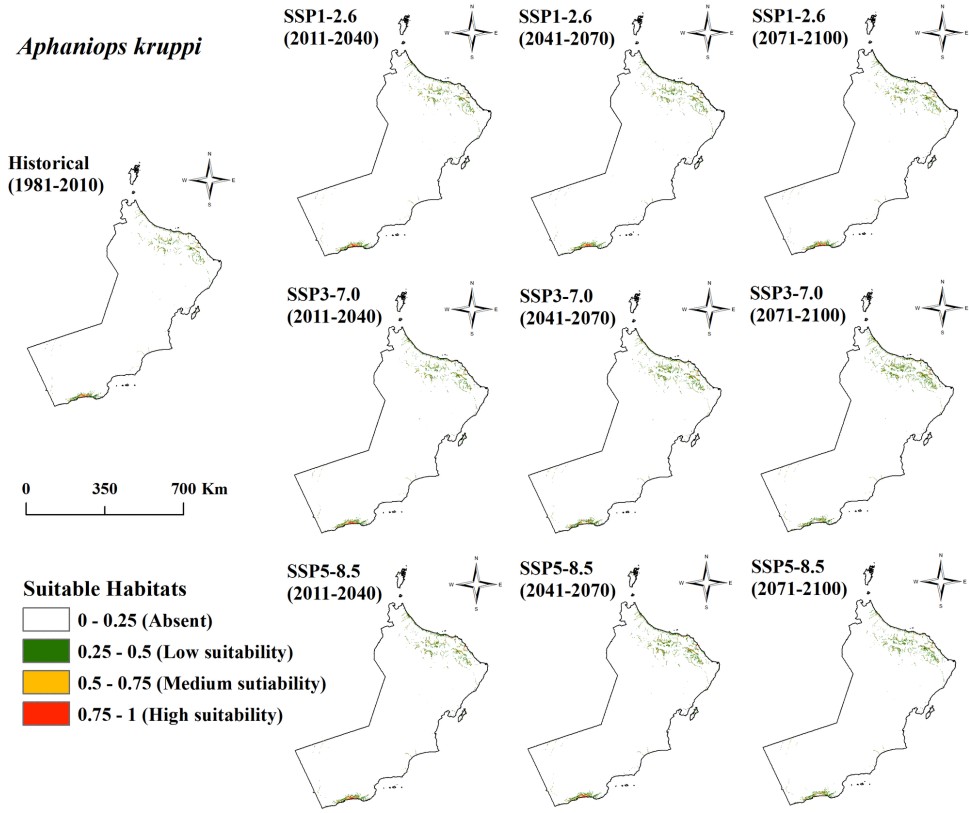

**Fig 4. Future habitat suitability for *A. kruppi* in Oman under SSP2.6/7.0/8.5 (1981-2100).**

discriminatory power while the TSS value of 0.667 validates excellent performance (S7 Fig). The Shapiro-Wilk test demonstrates that habitat suitability data follows normal distribution because W=0.904 and p=0.179 signify no significant statistical deviation (S7 Fig). The descriptive statistical analysis reveals an acceptable level of moderate positive skewness at 1.146 and platykurtic distribution with a kurtosis value of 1.103 for parametric analysis. The confusion matrix results show that classification accuracy stands at 83.3% with 10 correctly classified sites out of 12 total sites and includes 4 *true positives* (TP) and 6 *true negatives* (TN) while recording 2 *false negatives* (FN) and no *false positives* (FP). The confusion matrix analysis showed sensitivity (recall) at 66.7%, specificity at 100%, precision at 100% and F1 value at 80.0%.

**3.5.2. Habitat suitability and biodiversity trade-offs in stream ecosystems.** Correlation analyses between Habitat Suitability Index (HSI) and diversity metrics consistently indicated negative correlations across all alpha diversity metrics within the stream ecosystems studied (Table 4 and S7 Table). Bootstrap uncertainty quantification confirmed the robustness of these relationships: Shannon diversity index exhibited a moderate, negative correlation with HSI (r=−0.577, 95% CI: −0.840 to 0.009, p=0.049), as values varied from 0.678, at HSI of 0.496 in Wadi Darsait 3 (D3), to 0.034 at an HSI of 0.945 in Wadi Aday (A3), representing a 95% decrease across the HSI gradient. Evenness exhibited the strongest correlation (r=−0.589, 95% CI: −0.869 to −0.055, p=0.044), while Simpson's index demonstrated a negative trend with the confidence interval slightly crossing zero (r=−0.554, 95% CI: −0.849 to 0.011, p=0.062) (Fig 6).

Mean and standard deviation of the central tendency and variability of biodiversity metrics, based on the results of the bootstrap resampling for Shannon (mean±SD: 0.607±0.410, 95% CI: 0.377–0.829); Simpson (0.320±0.235, 95% CI: 0.188–0.456); and Evenness (0.451±0.296, 95% CI: 0.286–0.607) are presented in S8 Fig and S8 Table. The

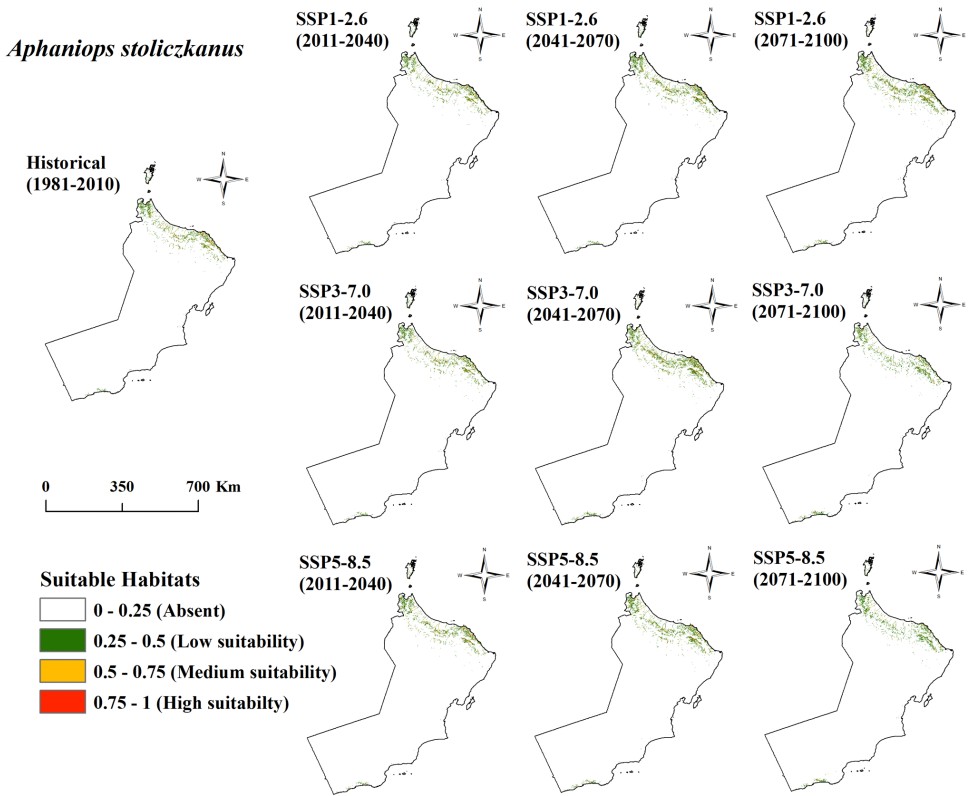

**Fig 5. Future habitat suitability for *A. stoliczkanus* in Oman under SSP2.6/7.0/8.5 (1981-2100).**

Leave-One-Out-Cross-Validation (LOOCV) showed moderate ability to transfer the HSI model to new sites ($R^2 = 0.852$, RMSE = 0.160, MAE = 0.133), an expected limitation due to the small number of study sites (n = 12) (S9 Fig) [73]. An evaluation of residuals of the model showed a systematic under-prediction bias (mean residual = 0.133) with the largest prediction error for site D3, A2, and K1 (residual = 0.323, 0.224, 0.223, respectively), indicating that these two wadis have unique ecological features not captured when the model is trained on other sites (S10 Fig and S9 Table).

The conclusions from the beta diversity analyses revealed clear spatial separation of stream communities based on their compositional differences (S10 Table). Moreover, the Bray-Curtis dissimilarity matrix [131] calculated the differences explicitly, finding relatively low dissimilarity values (0–0.143) between samples within the same stream grouping and higher dissimilarity values (and often exceeding 0.7) between stream groupings. For example, sites within group D displayed a high degree of similarity with each other (dissimilarity 0–0.143) but significantly dissimilarity to AW sites (dissimilarity 0.714–0.75). Across the AW group, dissimilarity was perfect (dissimilarity = 0), as it was for the K group.

The non-metric multidimensional scaling (NMDS) ordination analysis (Fig 6) revealed clear spatial segregation of stream communities with an excellent stress value of 0.044, indicating minimal distortion and high reliability of the 2D representation with little risk of misinterpretation [132]. AW sites clustered tightly in the upper right quadrant (NMDS 1 values = 0.543 to 1.025), indicating they shared similar community structure despite being different but physically separate sites, possibly indicating strong environmental filtering influences. K sites also clustered tightly in the lower right quadrant, indicating they exhibited similar community structure and respective environmental conditions or management practices (NMDS 1 values 0.370 to 0.661). A sites exhibited the most dispersion (NMDS 1 values −0.727 to 0.061), indicating that their community composition was more variable and could indicate that they are either exposed to increasingly diverse

**Table 3. Multivariate Environmental Similarity Surface (MESS) analysis for future climate scenarios.**

| Period | Climate Scenario | Species | MESS (Mean±SD) | Strong Extrapolation (%) | Moderate Extrapolation (%) | Analog Conditions (%) |
|---|---|---|---|---|---|---|
| 2011-2040 | SSP1–2.6 | *A. kruppi* | −35.61±79.59 | 85.45 | 7.67 | 6.87 |
| | | *A. stoliczkanus* | −18.01±70.14 | 62.07 | 26.46 | 11.47 |
| | SSP3–7.0 | *A. kruppi* | −33.35±78.28 | 85.26 | 8.08 | 6.66 |
| | | *A. stoliczkanus* | −17.55±69.94 | 63.31 | 25.74 | 10.95 |
| | SSP5–8.5 | *A. kruppi* | −36.28±78.27 | 87.26 | 6.24 | 6.5 |
| | | *A. stoliczkanus* | −18.38±69.94 | 66.59 | 23.15 | 10.26 |
| 2041-2070 | SSP1–2.6 | *A. kruppi* | −38.04±78.91 | 87.45 | 5.73 | 6.82 |
| | | *A. stoliczkanus* | −19.51±70.01 | 71.41 | 19.68 | 8.91 |
| | SSP3–7.0 | *A. kruppi* | −56.97±78.98 | 92.17 | 2.55 | 5.28 |
| | | *A. stoliczkanus* | −23.89±69.94 | 84.82 | 8.16 | 7.02 |
| | SSP5–8.5 | *A. kruppi* | −52.97±78.97 | 92.3 | 2.58 | 5.12 |
| | | *A. stoliczkanus* | −23.51±70 | 83.24 | 9.75 | 7.02 |
| 2071-2100 | SSP1–2.6 | *A. kruppi* | −52.94±78.99 | 92.29 | 2.62 | 5.09 |
| | | *A. stoliczkanus* | −23.75±70.02 | 83.24 | 9.69 | 7.07 |
| | SSP3–7.0 | *A. kruppi* | −84.79±79.33 | 94.66 | 0.63 | 4.72 |
| | | *A. stoliczkanus* | −35.82±69.82 | 91.56 | 2.65 | 5.79 |
| | SSP5–8.5 | *A. kruppi* | −92.77±80.82 | 94.92 | 0.38 | 4.7 |
| | | *A. stoliczkanus* | −39.75±70.07 | 92.64 | 1.82 | 5.53 |

**Table 4. Biodiversity and habitat suitability metrics for Hajar Mountain stream sites. HSI values, threshold classifications, and biodiversity metrics for 12 sites in Oman.**

| Stream ID | Shannon | Simpson | Evenness | HSI | HSI Threshold | Suitability Class |
|---|---|---|---|---|---|---|
| A1 | 1.213 | 0.665 | 0.875 | 0.584 | 0.51 ≤ HSI < 0.59 | Moderately Suitable |
| A2 | 0.871 | 0.441 | 0.629 | 0.490 | 0.47 ≤ HSI < 0.51 | Less Suitable |
| A3 | 0.034 | 0.010 | 0.031 | 0.945 | HSI ≥ 0.75 | Highly Suitable |
| AW1 | 1.184 | 0.660 | 0.854 | 0.587 | 0.59 ≤ HSI < 0.75 | Suitable |
| AW2 | 0.837 | 0.485 | 0.604 | 0.526 | 0.51 ≤ HSI < 0.59 | Moderately Suitable |
| AW3 | 0.942 | 0.501 | 0.680 | 0.443 | HSI < 0.47 | Unsuitable |
| D1 | 0.133 | 0.050 | 0.121 | 0.781 | HSI ≥ 0.75 | Highly Suitable |
| D2 | 0.185 | 0.076 | 0.134 | 0.581 | 0.51 ≤ HSI < 0.59 | Moderately Suitable |
| D3 | 0.678 | 0.381 | 0.617 | 0.496 | 0.47 ≤ HSI < 0.51 | Less Suitable |
| K1 | 0.234 | 0.095 | 0.169 | 0.635 | 0.59 ≤ HSI < 0.75 | Suitable |
| K2 | 0.533 | 0.268 | 0.384 | 0.724 | HSI ≥ 0.75 | Highly Suitable |
| K3 | 0.435 | 0.203 | 0.314 | 0.433 | HSI < 0.47 | Unsuitable |

Correlation analysis with bootstrap 95% CIs: HSI vs. Shannon: $r = −0.577$ (−0.858 to 0.037), $p = 0.049$; HSI vs. Evenness: $r = −0.589$ (−0.867 to −0.039), $p = 0.044$; HSI vs. Simpson: $r = −0.554$ (−0.840 to 0.024), $p = 0.062$. Cross-validation metrics: $R^2 = 0.003$, RMSE = 0.276, MAPE = 40.93%.

habitat conditions or potentially represent different stages of ecological succession. D sites clustered strongly in the negative NMDS 1 region (−1.096 to −0.866), indicating a level of uniqueness within their community composition and likely within another or different environment. Spatial interpolation (Fig 7) demonstrated an evident inverse geographic relationships between HSI (panel A) and diversity indices (panels B-D) during this study, where the areas of highest HSI (sites A3, D1, and K2) coincided with areas of lowest diversity, while sites A1, AW1, AW3, A2, and AW2 maintained higher diversity despite lower HSI values (Table 4).

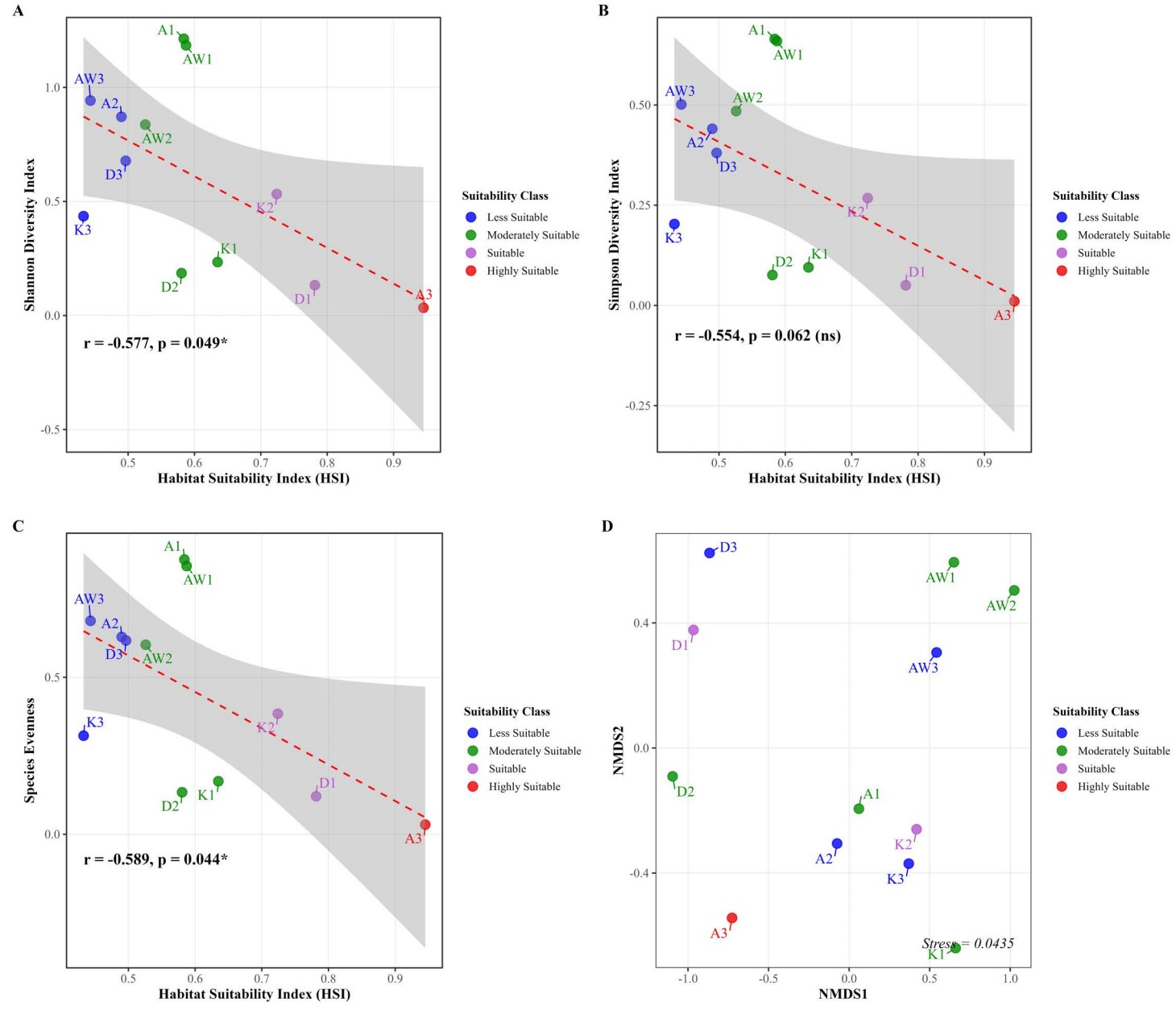

**Fig 6. Relationships between habitat suitability classes and aquatic biodiversity.** (A) Shannon Diversity Index (r = −0.577, p = 0.049), (B) Simpson's Diversity Index (r = −0.554, p = 0.062), (C) Evenness Index (r = −0.589, p = 0.044), and (D) NMDS ordination of community composition, colored by suitability class.

### 3.5.3. Environmental parameter importance.

An analysis of mean suitability values for the environment parameters showed a hierarchy of importance. In total, there were seven tiers of important variables. Dissolved oxygen (DO) had the highest importance with a mean suitability of 0.771±0.308, forming the first tier alone. The second tier included water velocity with a value of 0.729±0.287. Stream width and substrate texture shared equal importance in the third

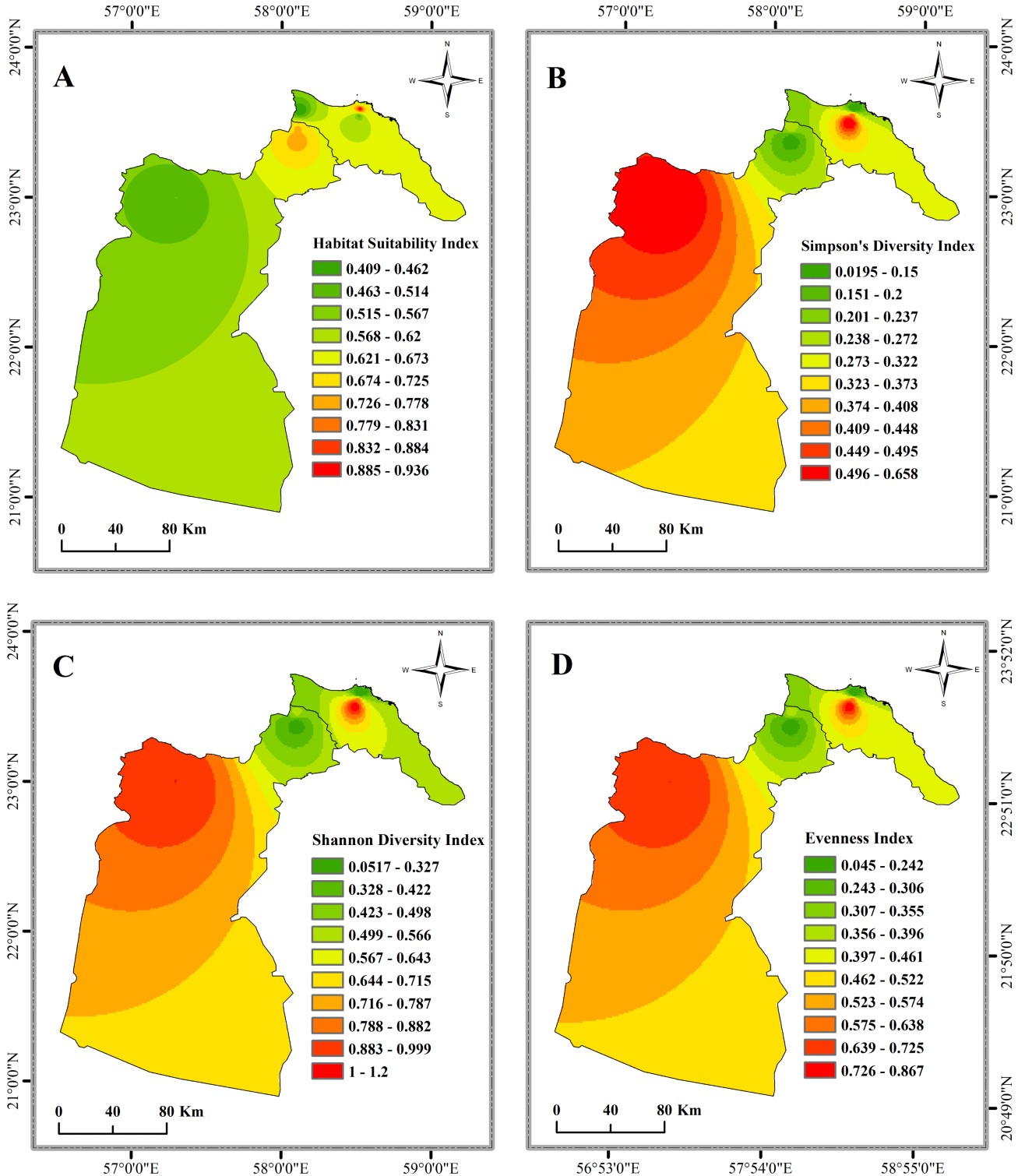

**Fig 7. Spatial distribution of Habitat Suitability Index and biodiversity metrics using Inverse Distance Weighting (IDW) interpolation.** (A) Habitat Suitability Index, (B) Shannon Diversity Index, (C) Simpson's Diversity Index, and (D) Evenness Index across the study region, demonstrating inverse geographic relationships between habitat suitability and biodiversity measures.

tier, with values of 0.719±0.297 and 0.719±0.247, respectively. Water temperature formed the fourth tier with a value of 0.707±0.354. pH comprised the fifth tier with a value of 0.694±0.345. BOD formed the sixth tier with a value of 0.686±0.309. The final tier included the water chemistry parameters: total dissolved solids (TDS), electrical conductivity (EC), salinity, depth, and turbidity, with values of 0.675±0.343, 0.675±0.343, 0.674±0.342, 0.673±0.286, and 0.672±0.292, respectively (Fig 8 and S11 Table).

### 3.5.4. Optimal habitat conditions.

The Optimal habitat parameter ranges for *Aphaniops* spp. were determined through examination of environmental measurements with Gaussian-derived suitability curves. *Aphaniops* spp. showed preference for certain ranges for all parameters. The physical habitat preferences were for intermediate depths (28.71–66.34 cm), intermediate stream width (3.63–8.62 m), and low to intermediate water velocity (0.04–0.24 m/s). Water chemistry parameters included optimal ranges found for temperature (29.29–30.78°C), slightly alkaline pH (8.07–8.50), moderate electrical conductivity (1164.46–2342.04 µS/cm), total dissolved solids (586.90–1187.57 mg/L), and salinity (0.57–1.18 ppt). The species preferred well-oxygenated water (DO: 6.42–8.66 mg/L) with moderate biochemical oxygen demand (2.83–3.93 mg/L) and low to intermediate turbidity (1.66–14.9 NTU) (Fig 9 and S12 Table).

### 3.5.5. Site-specific habitat suitability.

The Habitat Suitability Index (HSI) analysis conducted across the twelve study sites, using the quantile classification method, illustrated significant spatial variation in habitat quality (Table 4). Three sites (25.0%) fell into the "Highly Suitable" classification: Wadi Aday (A3) with the highest HSI, Wadi Darsait 1 (D1), and Wadi Fanja (K2), with values of 0.945, 0.781, and 0.724, respectively. In addition, these same sites were notably associated with different patterns of species diversity. Two sites (16.7%) were classified as "Suitable": Wadi Surur (K1) and Ain Wadhah 1 (AW1), with values of 0.635 and 0.587, respectively. Three sites (25.0%) had HSI values between 0.526 and

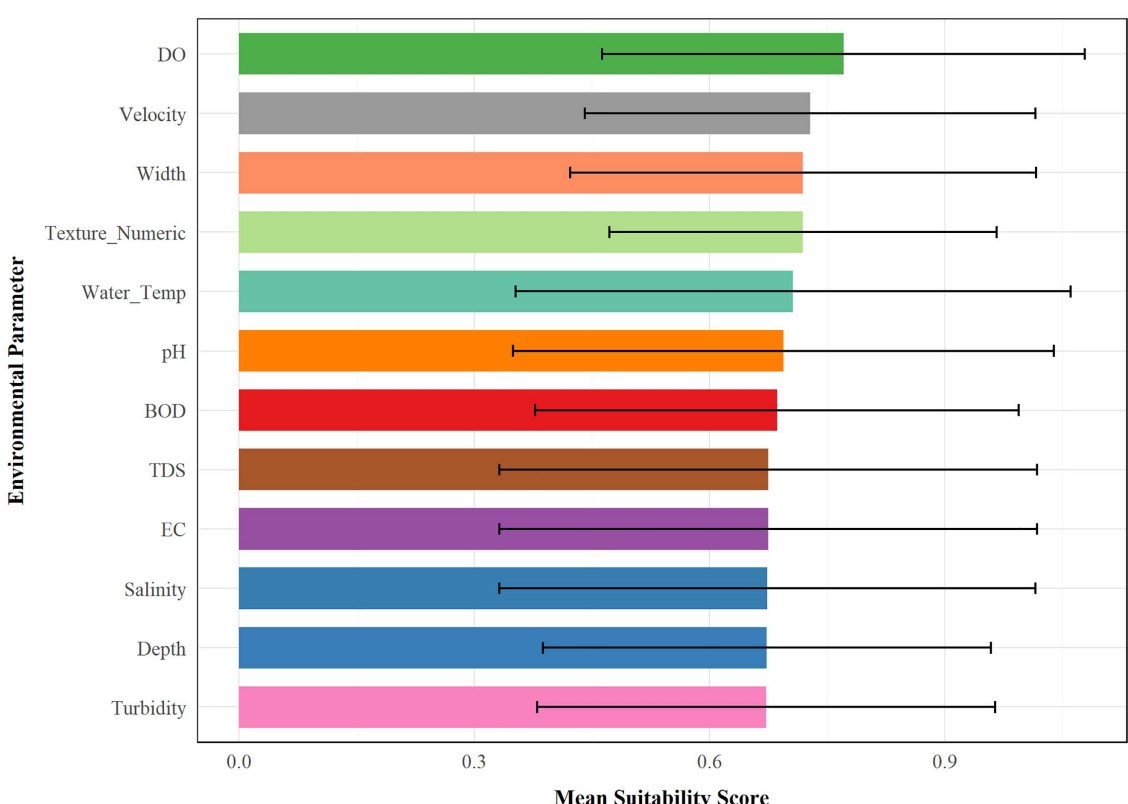

**Fig 8. Drivers of habitat suitability for *Aphaniops* spp.: Environmental variable contributions.**

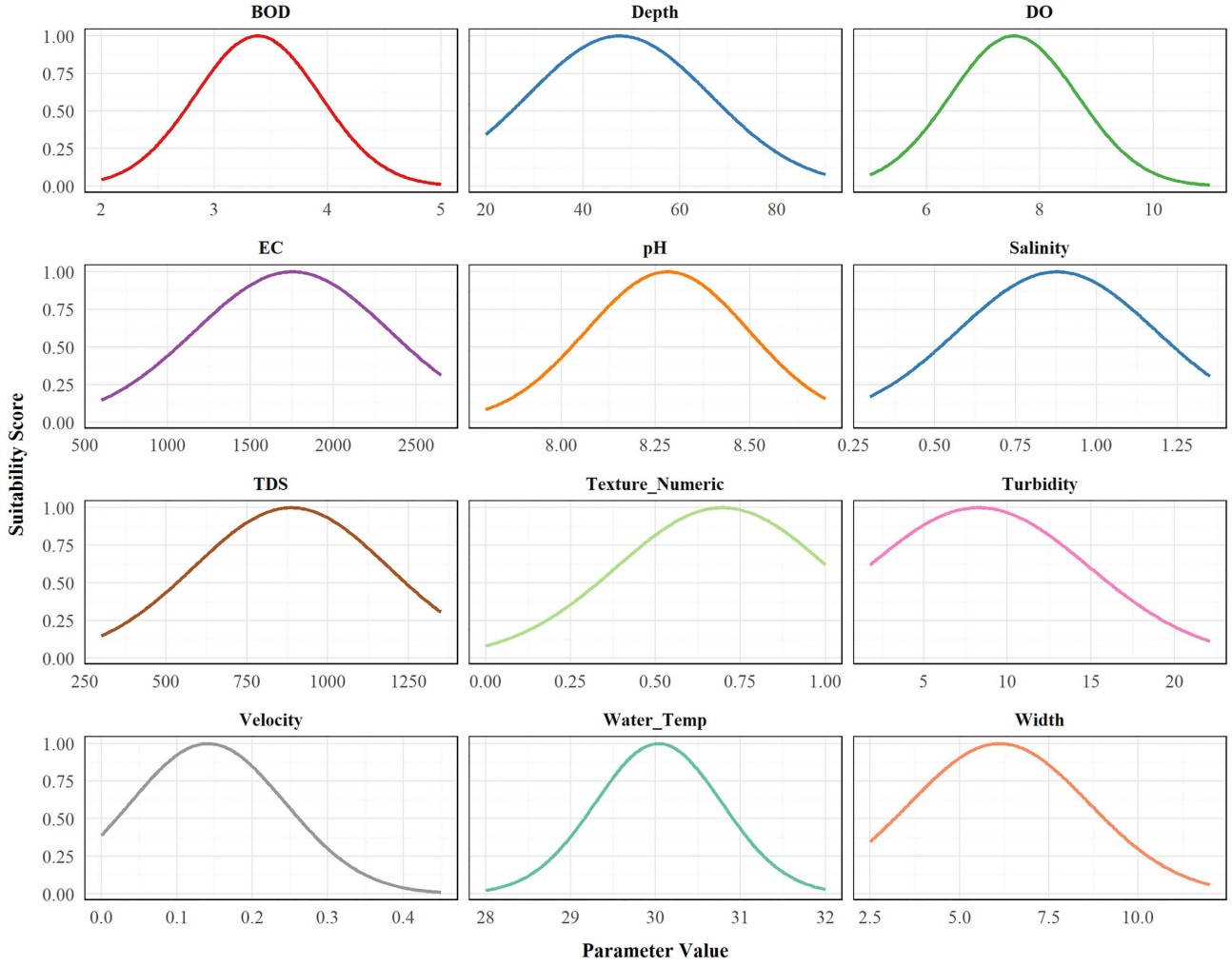

**Fig 9. Gaussian-fitted habitat suitability curves showing optimal environmental ranges for *Aphaniops* spp.**

0.584 and were classified as "Moderately Suitable": Ain Wadhah 2 (AW2), Wadi Darsait 2 (D2), and Wadi Al Amirat 1 (A1). Furthermore, two sites (16.7%) were classified as "Less Suitable": Wadi Darsait 3 (D3) and Wadi Al Amirat 2 (A2), with values of 0.490 and 0.496, respectively. Two sites (16.7%) were classified as "Unsuitable": Ain Wadhah 3 (AW3) and Wadi Al Khoud (K3), with values of 0.443 and 0.433, which had the lowest habitat quality conditions in the study area.

**3.5.6. Substrate texture impact assessment.** Analysis showed that HSI values with and without substrate texture varied across sites (Fig 10 and S13 Table). In total, substrate texture improved habitat suitability predictions for nine sites, with the largest improvements to HSI values at Ain Wadhah 3 (AW3; 6.7%), Wadi Al Khoud (K3; 6.1%), and Ain Wadhah 2 (AW2; 5.6%), respectively. The inclusion of texture also resulted in decreased HSI values at three sites, including Wadi Surur (K1), Wadi Fanja (K2), and Wadi Aday (A3), and their values were −17.0%, −1.4%, and −0.8%, respectively. This suggests a complex interaction between substrate characteristics and other environmental variables (Fig 10 and S13 Table).

Characterizations of the substrate composition patterns provide insight into the relationships that may exist between substrate texture and habitat suitability. The highest-suitability site, Wadi Aday (A3), with an HSI value of

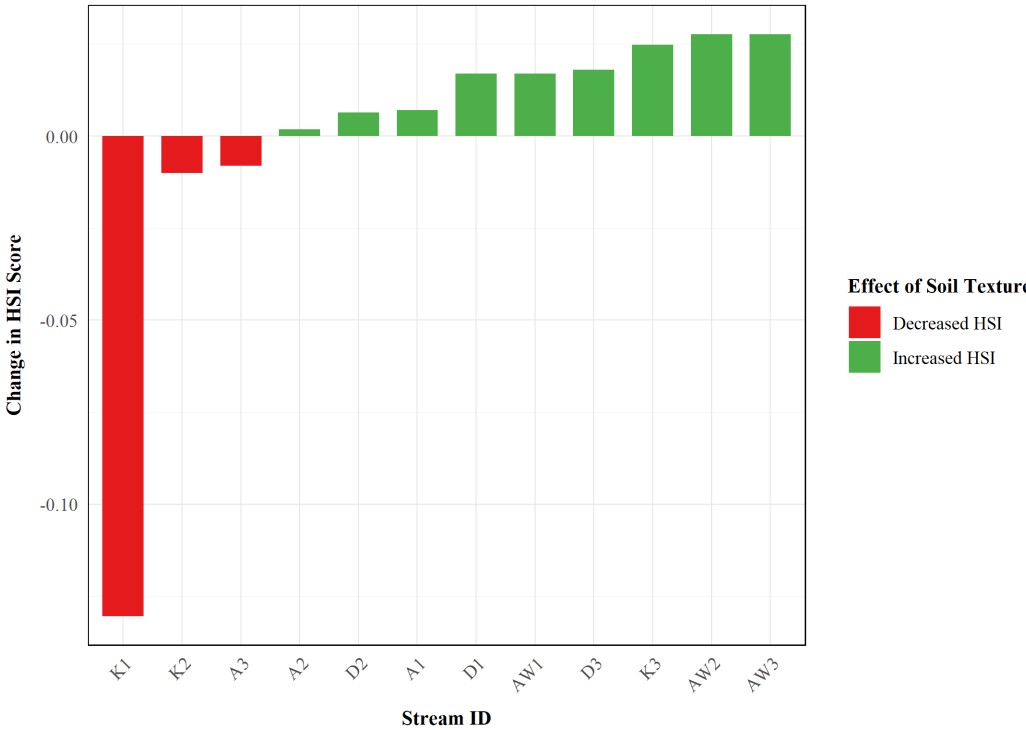

**Fig 10. Site-specific effects of substrate texture parameter inclusion on Habitat Suitability Index values.**

0.945, had a more balanced substrate composition of 47.45% sand and 48.1% gravel, with optimal amounts of each for *Aphaniops* spp. The sites with the largest improvements with texture inclusion had varied substrates: Ain Wadhah 3 (AW3) was characterized as gravel-dominant (60.44% gravel, 35.91% sand), Wadi Al Khoud (K3) as sand-dominant (80.62% sand, 12.47% gravel), and Ain Wadhah 2 (AW2) as largely gravel-dominant (90.49% gravel, 8.4% sand). In contrast, Wadi Surur (K1), which had the largest decrease (−17.0%) when substrate texture was included, exhibited a more varied substrate composition of 45.42% gravel, 35.23% sand, 14.26% silt, and 5.08% clay, resulting in a sandy loam texture. This indicates that more mixed substrate types might not be optimal for this specialized arid fish species (S2 Table).

### 3.6. Protected areas network analysis and stream conservation priorities

A spatial analysis of 31 of Oman's nature reserves boundaries obtained from the Environment Authority of Oman (designated by 2024) identifies areas of success and inadequacy in protecting freshwater habitats (Fig 11). The 31 protected areas total area is 17,441.45 km² and span multiple ecosystems that include coastal, desert and mountainous areas, representing 5.6% of the country's land area.

The conservation gap analysis (Table 5) revealed an extremely low level of protection for high suitability habitats for the two species analyzed; 0.31% for *A. kruppi* and 1.34% for *A. stoliczkanus* (S11 Fig). Progress toward Oman's national 2040 target (7.5% of habitat) is only 4.2% for *A. kruppi* and 17.8% for *A. stoliczkanus*, while progress toward the global CBD 30x30 target is even more limited (1.0% and 4.5%, respectively). This represents a severe gap in representation of the two species and demonstrates that a 24-fold increase in habitat protection will be required for *A. kruppi* and a six-fold increase for *A. stoliczkanus* to meet national conservation goals.

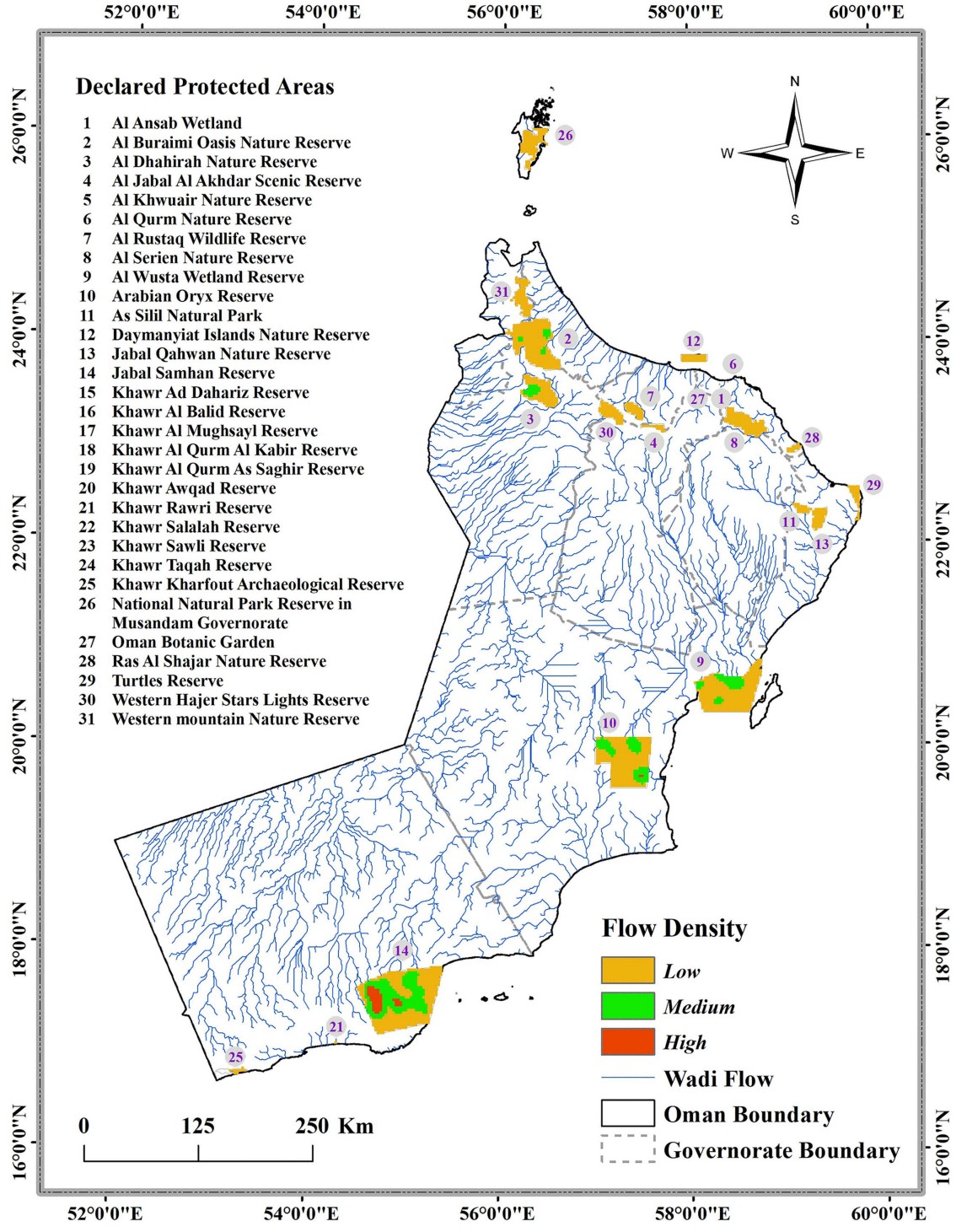

**Fig 11. Spatial analysis of Oman's protected areas and wadi flow density patterns for freshwater conservation prioritization.**

**Table 5. Conservation gaps and priority actions for killifish species in Oman. Protection status of high-suitability habitat (≥90th percentile) against national and international targets.**

| Conservation Metric | Aphaniops kruppi | Aphaniops stoliczkanus |
|---|---|---|
| **Current Protection Status** | | |
| 90th percentile suitability threshold | 0.0610 | 0.0842 |
| Current protection rate (%) | 0.31 | 1.34 |
| **Oman National Targets (2040) [a]** | | |
| Oman 2040 target (7.5% of habitat) | 7.5% | 7.5% |
| Progress toward Oman target (%) | 4.2 | 17.8 |
| Required increase to meet target | 24-fold | 6-fold |
| **CBD 30x30 Global Target** | | |
| CBD target (30% of habitat) | 30% | 30% |
| Progress toward CBD target (%) | 1.0 | 4.5 |
| Required increase to meet | 100-fold | 22-fold |
| **Wadi Network Protection Status [b]** | | |
| High-density wadis protected (%) | 2.6% (58 streams) | |
| Medium-density wadis protected (%) | 19.3% (427 streams) | |
| Low-density wadis protected (%) | 78.1% (1,731 streams) | |
| **Key protected areas (habitat km²)** | | |
| Al Buraimi Oasis Nature Reserve (PA 2) | 121.02 | 456.65 |
| Al Wusta Wetland Reserve (PA 9) | 401.8 | 0 |
| Al Serien Nature Reserve (PA 8) | 120.84 | 216.73 |
| Western Mountain Nature Reserve (PA 31) | 6.14 | 293.15 |
| Jabal Samhan Reserve (PA14) | 165.11 | 84.85 |
| National Natural Park Reserve in Musandam Governorate (PA 26) | 19.97 | 189.16 |
| Al Dhahirah Nature Reserve (PA 3) | 43.14 | 95.02 |
| Al Rustaq Wildlife Reserve (PA 7) | 77.49 | 57.51 |
| Ras Al Shajar Nature Reserve (PA 28) | 50.24 | 58.07 |
| Khor Kharfout Archaeological Reserve (PA 25) | 64 | 24.69 |
| **Priority Conservation Actions** | | |
| **Immediate priority** | Expand Jabal Samhan Reserve; protect monsoon-fed wadis in southern Oman. | Protect northern wadis; establish corridors linking PAs 31, 3, and 2. |
| **Key reserves for expansion** | PA 14, PA 9, PA 25, PA 7 | PA 2, PA 8, PA 31, PA 3, PA 28 |

[a] Oman's 2040 target is derived from its commitment to protect 25% of the CBD's 30x30 global goal for wetlands

[b] Wadi density protection targets: High: 50%; Medium: 30%; Low: 10%. Current status is shown as percentage of total streams within Protected Areas (PAs)

An examination of Oman's wadi networks also provides evidence of a greater degree of under-representation of these critical habitats. The flow density lines in the protected areas network are high density = 58 (2.6%), medium density = 427 (19.3%), low density = 1,731 (78.1%) (S12 Fig). This spatial analysis indicates that 97.4% of the high-density wadi systems fall outside of the currently designated protected areas, and numerous medium-to-high density wadi systems that are critical for freshwater are unprotected, especially where they cross areas of otherwise low-density or no-density development.

In addition to analyzing the collective PA network, the individual PA analyses identify priority reserves for each species (Fig 12 and Table 5). For example, for *A. kruppi*, the largest high-suitability habitats (165.11, 401.80, and 64.00 km²) are located in Jabal Samhan Reserve (PA 14), Al Wusta Wetland Reserve (PA 9), and Khor Kharfout Archaeological Reserve (PA 25); thus, these reserves are priority sites for expansion. For *A. stoliczkanus*, the largest high-suitability habitats

 

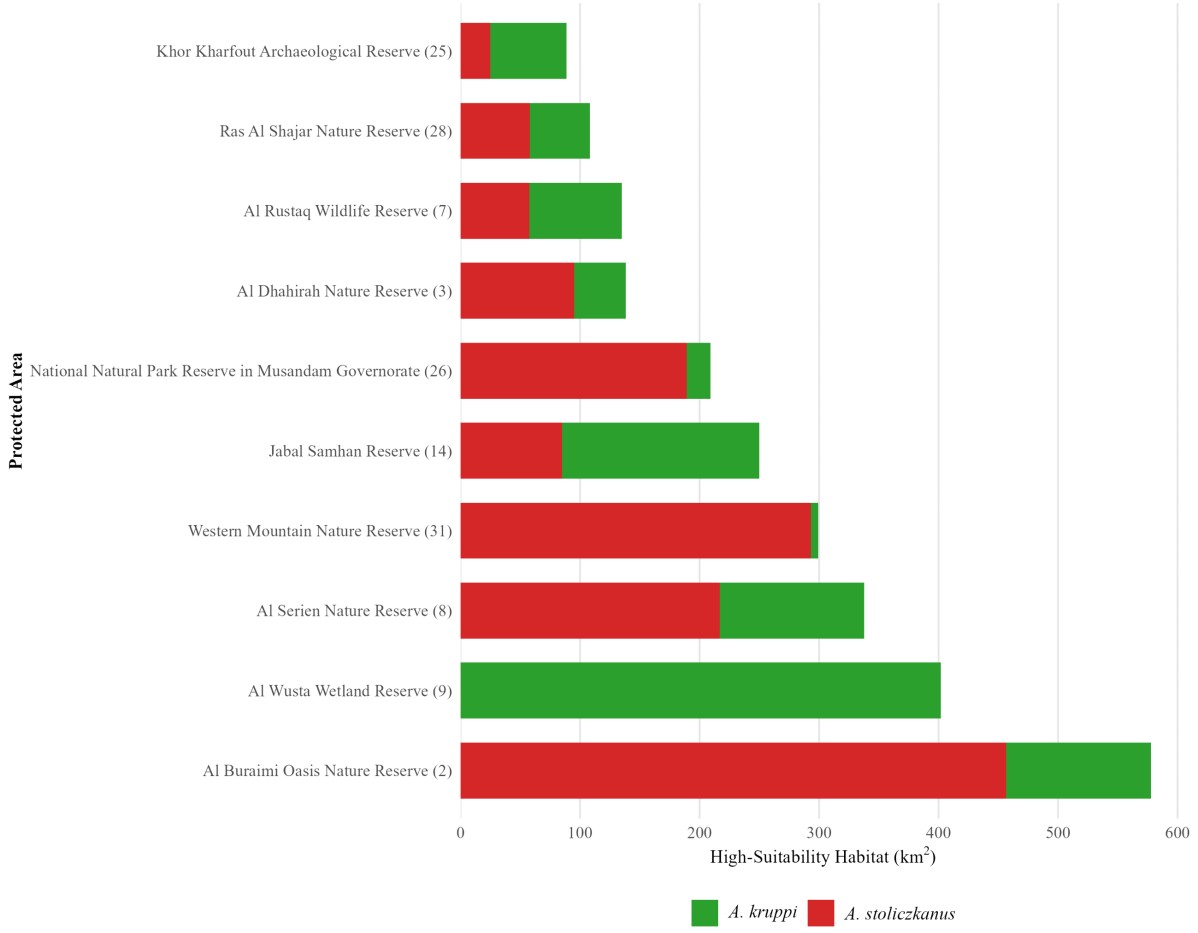

**Fig 12. Contribution of key protected areas to high-suitability habitat conservation.** Area (km²) of high-suitability habitat for *A. kruppi* and *A. stoliczkanus* within the top 10 protected areas in Oman.

(456.65, 216.73, and 293.15 km²) are found in Al Buraimi Oasis Nature Reserve (PA 2), Al Serien Nature Reserve (PA 8), and Western Mountain Nature Reserve (PA 31).

Immediate conservation priorities differ by species (Table 5). For example, immediate conservation priorities for *A. kruppi* include expansion of Jabal Samhan Reserve, as well as protection of monsoon-influenced southern wadis. Immediate conservation priorities for *A. stoliczkanus* include protection of northern winter-precipitation wadis and establishment of connectivity corridors between PAs 2, 3, and 31 to connect 844.82 km² of combined habitat that currently exists among those three reserves. Hybridization between the two species necessitates conservation efforts on a landscape level that maintain connectivity for adaptive gene flow.

## 4. Discussion

Oman's freshwater ecosystems support exceptional biodiversity within its arid landscape making it the top freshwater biodiversity location in the Arabian Peninsula [26]. *Aphaniops kruppi* and *Aphaniops stoliczkanus* represent endemic/native fish species in Oman that show unique spatial distribution patterns which demonstrate ecological niche partitioning that aligns with Oman's varied climate zones and water conditions. The study's comprehensive approaches which merge

species distribution modeling with habitat suitability assessment and protected area examination delivers essential information for managing conservation efforts in climate-threatened ecosystems.

## 4.1. Species distribution modeling of *A. kruppi* and *A. stoliczkanus*

The MaxEnt models demonstrated robust discriminatory performance for both *A. kruppi* (mean test AUC = 0.974 ± 0.014) and *A. stoliczkanus* (0.950 ± 0.018), while their TSS values (0.915 ± 0.090 and 0.817 ± 0.073, respectively) under MaxSS and MinROCdist thresholds [108,133,134] demonstrated their strong predictive capabilities. However, the default threshold (0.5) exhibited high specificity (0.990–0.993) but poor sensitivity (0.448–0.52), reflecting its unsuitability for rare species with extreme class imbalance (observed prevalence: 0.05–0.18%) [135,136].

Using MaxSS and MinROCdist thresholds improved sensitivity (0.904–0.973) and balanced accuracy (0.908–0.957), but generated prevalence over-predictions (5.9–9.9%) compared to the observed rate of <0.2% [103,137,138]. This prevalence mismatch is a known major concern when modeling rare species, as optimizing sensitivity (avoiding missed potential habitat areas) can result in a corresponding increase in the number of predicted total suitable area [103,136,137]. The larger prevalence mismatch for *A. stoliczkanus* (73 vs 57 flags for *A. kruppi*) may be due to either its broader ecological range or sampling bias or increased complexity when modeling a more widely distributed species across a range of environmental conditions [86,139]. These findings further emphasize the importance of utilizing multiple validation metrics and evaluating model predictions against the specific ecological characteristics and conservation objectives of the modeled species, rather than using only threshold-dependent metrics.

Our approach of model validation thoroughly tested all performance metrics using independent test data to ensure against over-fitting and followed well-established modeling standards [73,99]. This makes it especially notable that both species' models performed so well, since the smaller sample sizes are due to studying endemic species that have limited geographic ranges [75,76]. Additionally, the convergence of MaxSS and MinROCdist threshold values for *A. kruppi* at a single value, indicates a strong and robust consensus on an optimal cut-off between two different evaluation methods, thereby validating the process of selecting these threshold values [103]. The superior performance of *A. kruppi*'s model (higher TSS and AUC) may reflect narrower niche breadth or higher data resolution [55,140,141]. Conversely, the prevalence mismatches in *A. stoliczkanus* suggest broader environmental tolerance [55,86], underscoring that conservation planning requires threshold-dependent validation and context-specific metric selection [104,142]. This is especially critical for rare species where model results can be significantly impacted by minor data quality differences [95,143].

### 4.1.1. Patterns of biogeography and key environmental drivers.
Habitat suitability maps (Figs 4 and 5) reveal separate distribution patterns of the two *Aphaniops* species in Oman. *A. stoliczkanus* occupies a broad range along Oman's northern coastal areas within the Oman Mountain eco-region. While, the range of *A. kruppi* remains restricted to dispersed appropriate habitats along southern coastal areas associated with the Southwestern Arabian Coast eco-region.

The principal factors shaping *A. kruppi*'s restricted distribution pattern are the mean monthly climate moisture index (Cmi_m; 39.9%) and the mean diurnal air temperature range (Bio2; 18.3%). The Dhofar region's monsoon-dominated environment provides stable annual temperatures between 30–35°C [23] throughout summer rainfall periods which maintain essential moisture conditions vital for survival of this species. The species' stable presence within its limited range suggests it has specialized adaptations that enable it to thrive in the Dhofar region's stable climate conditions contrasting with Oman's interior plains where summer temperatures can reach over 50°C [23].In contrast, the broad distribution of *A. stoliczkanus* throughout northern Oman reflects its primary reliance on mean monthly precipitation during the coldest quarter (Bio19; 31%) which matches the winter precipitation trends from November to April found in the Al Hajar Mountains region of northern Oman [24,25]. The region receives up to 300 mm of annual rainfall which exceeds the precipitation levels found in interior regions (31 mm) and southern Oman (105 mm) [24]. *A. stoliczkanus* relies heavily on geomorphological factors such as the sediment transport index (STI; 20.2%) and stream power index (SPI; 13.3%)

because these factors likely connect with the many spring-fed wadis and streams originating from the Al Hajar Mountains where rainwater gathers in underground aquifers then surfaces as small lakes or creeks [60].

This climate-specific specialization reflects similar distribution patterns found in *Cyprinion muscatense* where seasonal precipitation emerged as a vital factor for species distribution under climate change [42], and *Garra shamal* research which showed essential interactions between precipitation patterns and stream sediment dynamics for defining suitable habitats [31].

**4.1.2. Projected impacts of climate change.** Future projections reveal species-specific vulnerabilities. *A. stoliczkanus* has broader yet unstable habitat patterns, with particular changes seen under SSP3–7.0 where low-suitability habitat grows from 0.02% to 1.04% in 2041–2070 before experiencing unexpected contractions. The medium-suitability habitat experiences erratic shifts with a 0.22% increase in 2041–2070 followed by a dramatic drop to −0.21% in 2071–2100. Later projections for high-suitability habitats show negative trends with reductions to −0.11% under SSP3–7.0 and −0.10% under SSP5–8.5.

For *A. kruppi*, this species experiences a rising trend in low-suitability habitat from 0.36% to 0.65% under SSP1–2.6 while medium-suitability zones moderately expand between 0.10% and 0.16%. The most dramatic shifts occur under SSP3–7.0, where low-suitability habitat increases reach 1.07% by 2071–2100.

The species' vulnerability might be connected to its reliance on winter precipitation trends which will become unpredictable due to climate change while the alterations in stream geomorphology might be caused by extreme weather events such as cyclones Gonu (2007), Phet (2010), and recent storms that brought extraordinary rainfall to the region [36]. Furthermore, *Aphaniops stoliczkanus* faces amplified natural vulnerabilities to climate change due to anthropogenic changes in Oman's freshwater ecosystems. In 2018, Oman established 155 groundwater and surface water dams [144,145] with many being located in northern areas where *A. stoliczkanus* populations live. New research findings demonstrate how dams disrupt river flow patterns and habitat connections [146] and predicts that dams will intensify warming effects because they block fish from moving to new areas as temperatures rise [147]. Beyond dams, freshwater systems face multiple threats including habitat destruction combined with high water extraction rates and pollution from domestic waste together with agricultural runoff and invasive species [10–12].

**4.1.3. Climate novelty and uncertainty assessment.** Our MESS analysis highlighted major differences in climate novelty exposure, a pattern similar to other freshwater species being impacted by climate change [148]. *A. kruppi* has an extrapolation risk that is much higher than *A. stoliczkanus* due to *A. kruppi* being distributed less widely and having a narrower range of environments it can tolerate. Since there are few analog conditions available, we have to be cautious when interpreting the climate projections, because all of the projected future climates represent new conditions relative to those used to train models [149]. However, even though the analog zones are small in number, they do represent possible climate refugia, which could potentially provide better predictability for the results of future habitat suitability projections [150], making it important to prioritize the conservation of these areas.

Additionally, the fact that the extrapolation risk for both species increases as the emission levels increase and the time period increases, indicates an urgent need for conservation actions, especially for *A. kruppi*, which exhibits near-complete strong extrapolation under the worst-case scenario, and highlights the vulnerable nature of freshwater specialists in the arid Arabian peninsula [30].

This research demonstrates that *Aphaniops stoliczkanus* and *A. kruppi* display separate spatial distribution patterns because of their unique adaptations to local climate conditions. Conservation strategies must be tailored to each species' specific needs: the conservation of *A. kruppi* relies on the preservation of moisture regimes in southern wadi systems due to monsoon pattern changes while *A. stoliczkanus* requires maintenance of natural flow regimes and sediment dynamics because of its sensitivity to stream power and sediment transport indices [31]. Conservation plans must focus on the specific hydrogeomorphological requirements of endemic fishes to protect them against climate change and human-induced habitat changes. These unique conservation needs of these species demonstrate why targeted strategies are crucial

instead of generalized methods in the Arabian Peninsula due to growing aridity affecting aquatic biodiversity in temporary and permanent water channels [30,42].

## 4.2. Habitat suitability assessment of *Aphaniops* spp. in 12 Hajar Mountain streams

### 4.2.1. Model performance and HSI distribution.

The habitat suitability model developed for *Aphaniops* spp. demonstrated Excellent predictive performance, as detailed in Section 3.2.1. Furthermore, promoted model reliability, with these values similar to or greater than other studies modeling fish habitat in freshwater [108,151], consistent with recommended approaches given the data-limited nature of the study [76,152]. Our validation process uncovered some nuance to the results; the HSI model had good descriptive validity for our sampling sites, but LOOCV indicated that the HSI model is fairly robust ($R^2 = 0.852$) but with moderately large errors (RMSE = 0.160, MAE = 0.133). We characterized this as "moderate transferability" because it is difficult to generalize an ecological model as complex as ours when trained using such a small sample size (n = 12) [73]. The results of the residual analysis suggested a systematic under-prediction bias (mean residual = 0.133) with the greatest prediction errors at sites D2 (0.343), A2 (0.224), and K1 (0.223), indicating that these wadis may possess unique environmental or habitat characteristics which were not adequately captured by the broader dataset (S9 Table).

The characteristics assigned to *Aphaniops* spp. as a taxonomic category, and that fact that hybridization has been documented between *A. stoliczkanus* and *A. kruppi* in northern Oman wadis with mixed mitochondrial haplotypes suggesting gene flow, further complicates conservation and taxonomy at distinct species levels [153,154]. The hybridization could facilitate adaptive introgression as well as climate resilience in species independently experiencing evolutionary pressures through the movement of adaptive alleles [155,156]. Although our model's classifications were conservative as evidenced by our classification metric results, the bootstrap validation confirms that our fundamental ecological conclusion that there is an inverse relationship between HSI and biodiversity was still statistically valid; validating the specialization-dominance trade-off regardless of the previously mentioned predictive limitations [157,158]. The observed habitat suitability distribution classes across our study sites reflects an ecological reality that optimal conditions are generally rarer than suboptimal conditions [159,160].

### 4.2.2. Habitat suitability and biodiversity trade-offs in stream ecosystems.

This study demonstrates a counter-intuitive, inverse relationship between habitat suitability for desert killifish *Aphaniops* spp. and overall aquatic biodiversity within stream ecosystems (Table 4 and Fig 7). This finding is robustly supported through bootstrap validation of negative correlation of HSI to each of the alpha diversity metrics. This challenges the prevailing belief that more favorable habitat conditions result in increased biodiversity. Our findings reveal that a desert killifish *Aphaniops* spp. can thrive in highly-specialized habitat conditions that do not support overall high aquatic biodiversity, as expected from several other desert killifish species [161–163]. Instead, our observed model illustrates a typical specialization-dominance trade-off, in which habitat specialists can achieve numerical dominance in harsh environmental conditions, but only at the cost of community diversity [164,165]. In high HSI sites, *Aphaniops* spp. is likely acting as a specialist, where evolutionary adaption may enable them not only to exploit narrow environmental conditions but also to asymmetrically exclude generalist species that require broader niche breadths [166].

The negative relationships between HSI and biodiversity metrics (Table 4) indicate that as habitats become more suitable for *Aphaniops* spp., they become less suitable to other species and therefore the community becomes simpler with one dominant specialist [157,158]. This pattern aligns with habitat filtering theory, with harsh environmental conditions acting as strong ecological filters that differentially select for specialization with traits that filter out species that lack those traits [167,168]. Moreover, these trends may be explained by the Intermediate Disturbance Hypothesis [169] and competitive exclusion principles [170]. Furthermore, the pattern observed resonates with broader ecological theory depicting that specialist species are competitively dominant within their preferred habitats and develop conditions conducive to their own dominance due to positive-feedback mechanisms [171–173]. In extreme environments, like desert streams, environmental filtering dominates the prevailing assembly mechanisms and trait convergence for specialization leads to lower functional

diversity [174–176]. This is intensified in aquatic desert ecosystems, with physiological convergence forcing even stronger selection where environmental filtering takes the form of temperature tolerances, oxygen availability, and osmotic regulation [176–179].

The spatial patterns illustrated in the NMDS analysis (as detailed in section 4.2.3) add further support for the specialization-dominance framework, it also demonstrates that environmental filtering operates differently across the different landscape contexts of our study area (Fig 6).

The spatial pattern of HSI values and diversity metrics across our study area provide additional evidence supporting our theoretical prediction. This spatial pattern indicates that the specialization-dominance trade-off operates at landscape scales, and specialist dominated communities form discrete areas in the metacommunity [180–182]. Of note, even though AW3 indicated a small improvement in habitat suitability (+6.7%) when substrate texture was considered in the model, the site still remained an unsuitable habitat (HSI 0.443). The flash flood event that changed AW3 flowing stream into isolated, inundated pools (recorded during the study period), created conditions that were some of the least suitable for *Aphaniops* spp., suggesting that a flood may create suboptimal habitat conditions even with some localized substrate improvements [183,184]. The breakdown in flow is a significant disturbance that irrevocably changes the physical template that shapes fish habitat quality and community structure [185, 186]. Specifically, pool isolation usually decreases habitat connectivity and alters the availability of resources for specialized stream fishes [187,188]. Such disturbances may disrupt the dominance of specialists temporarily, and potentially allow generalist species to colonize, as our results indicate that *Aphaniops* spp. is able to maintain its competitive superiority even under moderate disturbance, indicating its remarkable adaptation to extreme desert stream conditions [189,190].

**4.2.3. Environmental parameter hierarchy.** The hierarchical analyses of environmental parameter presented in section 3.2.3 identified dissolved oxygen as the most influential parameter in determining habitat suitability, a finding consistent with the physiological requirements of cyprinodontiform fishes. The importance of dissolved oxygen for *Aphaniops* spp. is likely a common occurrence among cyprinodontiformes fishes, and the optimal dissolved oxygen levels of 6.42–8.66 mg/L would be similar to their related species [191]. Most freshwater fish require 5–9 mg/L DO to thrive, with levels below 5 mg/L inducing stress and concentrations under 2 mg/L proving fatal [192].

The preference for moderate to slow current velocities identified in our analysis has important implications for understanding killifish habitat selection. This suggests that the habitat preferred by *Aphaniops* species probably exists in springs, small streams and the margins of shallow lakes, where current velocities are generally low [27,193,194]. The norm also suggests that killifish prefer to occupy habitats with low or no water movement, which is supported by many species being adapted to lentic or slow habitats of sufficient depth and volume [195]. While stream width and substrate emerged as equally important parameters in our model, quantitative relationships between stream width and killifish population density remain poorly understood, representing a critical research gap requiring field studies combining hydromorphological surveys with population assessments in arid environments. This context-dependent importance of habitat variables aligns with hierarchical habitat selection theory [196,197].

The thermal tolerance range reported in this study demonstrates the evolutionary specialization of killifish in arid environments. Thermal specialization of the killifish is represented in our findings, and additional studies on related Iranian *Aphanius* species also support findings with more narrow tolerances [27,193,194]. Moreover, this optimal temperature range aligns with thermal preferences documented for related cyprinodontoid species in arid environments [161,198]. *Aphaniops* spp. have the ability to thrive in alkaline conditions, which requires specialized physiological adaptations. This indicates specialized physiological adaptations and adaptations that facilitate alkaline survival under conditions that would normally disturb embryonic development [199], and necessitate alterations to the gill morphology to manage extreme conditions [200]. Alkaline tolerance indicates evolutionary adaptations to arid systems [201].

Although water quality characteristics (EC, TDS, salinity) were moderately important to our model, they have been shown to be highly flexible by the extreme euryhaline abilities of Arabian Peninsula Killifish. Arabian Peninsula killifish

show outstanding euryhaline abilities, with individuals tolerating extreme salinity ranges from freshwater oases (0.7–1.5 ppt) to hypersaline coastal lagoons (41–44 ppt), likely through specialized molecular adaptations and mechanisms [202]. Likewise, the level of tolerance we observed for turbidity will influence the ability of the killifish to forage as they move through different environments. Yet, visual detection and predation efficiency in closely related species decreases significantly above 10 FTU (Formazin Turbidity Units), with 32% reduction in feeding efficiency under algal turbidity compared to clear water [203]. These adaptions allowed them to colonize variable ephemeral pools and streams throughout the Arabian Peninsula [195]. While other water chemistry adaptations vary substantially for endemic species in the eco-sensitive habitats in the Arabian Peninsula; especially considering compounding effects of climate change, groundwater exploitation, agriculture runoff, invasive species, and pollution [12,30,204].

The complex role of substrate texture in our model, with both positive and negative effects at different sites, demonstrates flexible substrate requirements rather than absolute substrate preference. Overall, results suggested that *Aphaniops* spp. displayed substrate rather than requiring specific compositions. While, the highest-suitability site (A3, HSI: 0.945) noted optimal habitat condition with balanced substrate composition; (47.45% sand, 48.1% gravel). Sites that exhibited improvements with the inclusion of substrate texture exhibited different profiles; AW3 was highly gravel-dominated (60.44% gravel), K3 was highly-sand dominated (80.62% sand), while AW2 was highly gravel-dominated (90.49% gravel). Such substrate flexibility among substrate composition range has been reported among other killifish species [29,205,206].

**4.2.4. Community assembly patterns (NMDS Analysis).** The patterns of specialization-dominance articulated in section 4.2.2 and their reference to NMDS suggest how these ecological mechanisms shift spatially across landscape contexts, with environmental filtering emphasizing different degrees of intensity dependent on local constraints. The NMDS analysis effectively evaluated community assembly mechanisms and provided an excellent stress value of 0.044, thus yielding a high level of confidence that the observed clustering was a true ecological difference rather than an analytical artifact [207,208]. This excellent fit allows for meaningful interpretation of spatial segregation patterns as corresponding to underlying community assembly mechanisms [209]. The spatial segregation evidently resulted in the identification of ecological clustering that reflected deterministic community assembly processes that varied by spatial scale [210], with environmental filtering being the most prevalent mechanism structuring communities within different landscape contexts [167]. The Site-specific community assembly patterns were as follows and summarized in (S14 Table):

- Ain Wadhah (AW) sites representing 'Agricultural-Woodland Interface': These sites clustered tightly on the upper right quadrant (NMDS1 values 0.543–1.025), demonstrating strong geological and topographic environmental filtering effects associated with mountain-stream interface landscapes [211]. The tight clustering indicates mountain-influenced hydrology and geochemistry where distinctive environmental conditions promote specialized community assemblages. Importantly, site AW3 underwent significant temporal changes over the study, changing from a connected flowing stream to isolated pools due to the effects of flash flooding. This resulted in its ecological characteristics altering while coexisting with its montane environmental condition. Overall, the homogeneous clustering suggests montane environmental filtering creates consistent selective pressures favoring species adapted to oligotrophic habitats, warm but thermally stable waterways, and specific mineral compositions based on the geology of mountains [167]. It is estimated that the mountain environment filtering selects for oligotrophic specialists and species that are adapted to low-conductivity, low-salinity mountain spring waters [212,213]. The unique hydrogeochemical signatures of mountain groundwater systems is due to how the surface water interacts with geological features [214], whereby the chemical evolution of groundwater is the result of the weathering of silicate and carbonate minerals [215]. These characteristics are expected to strongly influence downstream aquatic communities through mountain influences hydrogeochemical processes.

- Al Khoud (K) sites representing a 'Traditional Wadi Systems': These sites exhibited a tight clustering in the lower right quadrant (NMDS1 values 0.370–0.661), revealing mixed urban-agricultural-wadi landscape patterns similar across the

entirety of the Ad Dakhiliyah region. This pattern indicates similar approximately moderate anthropogenic influence with similar arid lowland geology and wadi geomorphology [216–218]. These sites are representative of a moderate disturbance regime, and show settled-agricultural interface impacts including moderate urban engagement and palm cultivation, delimiting the predictable environmental filtering conditions but buffered by the natural wadi geomorphology [219,220]. This environmental filtering selects communities adapted to approximately moderate anthropogenic stress being shaped by semi-arid wadi hydrology, as well as geological constraints, and is in accordance with urban stream ecology studies showing that moderate levels of anthropogenic influence can lead to stable community compositions that are buffered by natural geomorphological features [221,222].

- Al Amirat (A) sites representing an 'Urban Gradient Sites': These sites exhibited the most spread across NMDS1 (−0.727 to 0.061), reflecting areas of abundant urban disturbance variability across Wadi Al Amirat and Wadi Aday systems. The on-site characteristics indicate a reasonable urbanization gradient: A1 = urbanization in a moderate state; A2 = mountain wadi ecosystems, fairly preserved; and A3 = developed heavily. The dispersion indicates a complex interaction between deterministic environmental filtering and stochastic assembly processes [223], whereby urbanization-induced habitat heterogeneity allows the assemblages to be constructed, in part, by the creation of microenvironmental mosaics [224,225]. The pattern of variability represents how urban gradients can result in complex community assembly dynamics in geographic regions [226], and once again, the same urban disturbance variability in levels overrides habitat similarities within the region to create new selective environments across different intensities of disturbance from moderate development to preserved systems to urban stream syndrome conditions [221,227].

- Darsait (D) sites representing a 'Complete Urban Wadi System': These sites formed a clearly identifiable grouping in the negative NMDS1 region (−1.096 to −0.866), reflecting their unique position as a complete wadi-to-sea continuum within Wadi Darsait system. These sites represent different watershed positions: D1 (urban headwaters), D2 (middle reaches with sewage treatment), and D3 (coastal discharge with marine influence). The tight clustering indicated that regional-scale environmental filtering and shared watershed identity had created a unique regional pool of species that was substantially different from the remaining wadi systems. This pattern reflects the combined influence of urban stream syndrome effects – characterized by flashier hydrographs, elevated nutrients, altered channel morphology, and reduced biotic richness with tolerant species dominance [220,221,228] alongside longitudinal river continuum dynamics [229,230] and unique anthropogenic influences including disruption of longitudinal connectivity [231,232] and wastewater treatment impacts [228,233] and marine connectivity [234] that create environmental conditions not found in other watershed groups.

   **4.2.5. Beta diversity and community turnover.** Patterns of beta diversity, measured using Bray-Curtis dissimilarity analysis [131], help to show how habitat suitability-diversity trade-offs occur across the landscape. Generally speaking, the patterns reflect the fundamental area-heterogeneity trade-off; increasing environmental heterogeneity allows suitable conditions for more species, while consequently decreasing the effective area available for individual species [235]. The within-group homogeneity (dissimilarity values of 0–0.143) was starkly different from the between-group heterogeneity (often over 0.7), which indicate there was strong community turnover along environmental gradients [236]. The perfect within-group similarity observed in the AW and K groups (dissimilarity = 0), indicates environmental filtering may function as an important assembly mechanism locally [237]. Alternatively, we found significant dissimilarity between D and AW sites (0.714–0.75), which suggests there was strong turnover between the different watershed systems, and shows that species sorting mechanisms were operating at regional scales [210]. These beta diversity patterns support the NMDS results, indicating that the specialization-dominance tradeoff is an aspect of metacommunities, in which community assemblies are formed based upon the environmental filtering at the landscape scale.

### 4.3. Protected areas network analysis and conservation planning implications

Oman has 31 reserves totaling 17,441.45 km² in protected area coverage by 2024. Unfortunately, this protected area network (PA) is inadequate to capture the freshwater habitats of Oman for endemic fish such as *Aphaniops kruppi* and native *A. stoliczkanus*, especially given the spatial analysis results of earlier studies, which reported significant misrepresentation of wadi systems within the protected area network—which also applies to *Cyprinion muscatense* [42]. Our analysis reveals substantial difference between the location of PAs and the pattern of occurrence of the most important habitats for both *A. kruppi* and *A. stoliczkanus*, specifically the wadi systems that serve as critical habitat for these two species.

**4.3.1. Quantifying the protection gap.** The scale of this mismatch is quantified in Table 5. The amount of high-suitable habitat is extremely small (0.31–1.34%) for both species and represents an extremely small percent of Oman's goal of protecting 7.5% of all critical wetlands by 2040. Therefore, the total amount of critical habitat to be protected to meet the national goal is 24-folds for *A. kruppi* and 6-folds for *A. stoliczkanus* higher than the amounts currently being protected. Furthermore, there is an even larger deficit regarding the wadi corridor needed to maintain the populations of our fish species. As shown in S12 Fig, only 2.6% of the high-density wadis are currently located within the boundaries of PAs when it is estimated that 50% of the wadis need to be included in PAs in order to preserve the necessary connectivity for metapopulations. Therefore, approximately 97.4% of the wadis are unprotected. This represents a significant gap in planning for conservation as the linear nature of the wadis are vital for metapopulation persistence and genetic connectivity in fragmented landscapes [238,239].

**4.3.2. Key priority areas for conservation action.** Our results identify specific PAs that represent the current core of protected habitat for each species. These should be targeted for increased management and potential strategic expansion (see Fig 12). For *A. kruppi*, the Jabal Samhan Reserve (PA 14), Al Wusta Wetland Reserve (PA 9), and Khor Kharfout Archaeological Reserve (PA 25) reserves are most important. For *A. stoliczkanus*, the Al Buraimi Oasis Nature Reserve (PA 2), Al Serien Nature Reserve (PA 8), and Western Mountain Nature Reserve (PA 31) reserves are most significant. Enhancing the management of these reserves and strategically expanding them will provide the greatest efficiency in advancing national conservation goals.

**4.3.3. Policy integration and future challenges.** The findings of this study align with global trends where freshwater ecosystems are consistently underrepresented in protected area networks [5]. The identification of priority areas and the immediate need to protect wadi corridors directly support Oman Vision 2040's strategic direction of achieving "Effective, Balanced and Resilient Ecosystems" [128]. However, achieving this goal will require confronting compound threats such as climate change altering precipitation regimes upon which these species depend [22,24] and the increasing threats from humans, such as dams, extraction of freshwater, invasive species and agricultural runoff, and pollution threatening unprotected wadis, especially in central Oman [12,19,240].

Therefore, future conservation must include comprehensive, landscape-level strategies. Moreover, the described hybridization between *A. stoliczkanus* and *A. kruppi* necessitates a shift from species to landscape-level conservation planning that protects the whole *Aphaniops* species complex while retaining connectivity to enable adaptive gene flow and hybrid zones as potential sources of evolutionary change [241,242]. Ultimately, conservation for Oman's endemic fish species will require a complicated and multi-facetted approach that maximizes the potential to increase protections for the critical wadi corridors [30], carefully manage freshwater within current protected areas, and develop buffer zones around key sites [51] to ensure the long-term existence of Oman's unique freshwater biodiversity.

### 4.4. Study limitations

We acknowledge that the sample size used to assess *A. kruppi* (n = 40) is a limitation, it is an accurate representation of the difficulty associated with studying endemic species due to their limited geographic distribution [75]. As the observed

prevalence rates of *A. kruppi* were below 0.2%, this also adds to the difficulty of selecting thresholds and predicting habitat within the context of species distribution models [136,137]. The prevalence mismatches under sensitive thresholds reflect this, a well-known challenge when modeling rare species with extreme class imbalance [136,137]. Although we applied 1 km spatial filters to reduce autocorrelation between samples [72], the small sample size did not allow us to employ more complex spatial blocking techniques to evaluate our species distribution models [73]. However, we utilized recommended standards for model evaluation [99] and used independent test data and were transparent regarding the limitations of our methodology.

The integrated approach encompasses all aspects of assessing habitat suitability; however, the Habitat Suitability Index (HSI) model demonstrated several limitations as an assessment tool. The HSI model demonstrated strong cross validation fit and had very good descriptive validity, nonetheless, it systematically under-predicted habitat suitability values. Prediction errors were highest at specific sites indicating that local habitat conditions or unmeasured micro-habitat variables were not fully captured by the model. This limitation of habitat models based on the local ecological conditions is also reflected in the limited predictive power of these models when there are low numbers of study sites sampled over broad environmental ranges. Therefore, future studies would benefit from increasing the number of sampling locations sampled across a variety of different environmental ranges to increase the generalizability and predictive ability of the habitat models for use in developing conservation plans.

While this integrated approach is comprehensive, there are limitations that need to be acknowledged. Our species occurrence data represent moments in time which likely do not consider the seasonal dynamics of desert wadi systems. Extreme hydrological variability between wet and dry periods can greatly influence species distributions as well as the available habitat for species [243–245]. We also have to consider spatial sampling bias as accessibility may have limited our sampling in mountainous terrain and remote wadis, resulting in spatial bias for geographic areas that were better surveyed, and potentially influencing model predictions [92,246]. Our climate change projections are based on a single global climate model (GFDL-ESM4) which has internally consistent projections however it doesn't contain all the uncertainty associated with using a multi-model ensemble for climate projection modeling. Although we rigorously quantified this uncertainty through bootstrapping and MESS analysis, which revealed significant extrapolation risk (particularly for *A. kruppi*), future research can further reduce these limitations by using ensemble approaches that include multiple GCMs to better define the uncertainty in climate projections and to generate a more robust estimate of the possible range shift of species under various climate change scenarios.

The hybridization we have reported between *A. kruppi* and *A. stoliczkanus* denotes taxonomic uncertainty surrounding species identification, particularly in northern populations with mixed mitochondrial haplotypes presumed to be undergoing gene flow, probably impacting traditional conservation strategies at the species level [153,154]. The process of integrating these landscape-scale models with site-based habitat evaluations requires reconciliations of spatial resolutions, with coarse-scale bioclimatic variables potentially omitting finer-scale habitat features that may help the continued persistence of species [40]. Climate projection uncertainties remain acute for arid regions, as global climate models illustrate variabilities in predictions of precipitation and temperature, limiting confidence in habitat projections [32,33]. As a result, our MESS analysis suggests that most projected future conditions, especially those under high-emission scenarios, represent novel conditions outside our models' current training domain, necessitating cautious interpretation.

Our protected area gap assessment provides a number of important insights, however there are many limitations. The conservation targets we employed were derived from Oman's National Environment Strategy indicator for wetland conservation ("Percentage of beneficiary protected areas [Ramsar sites] of total wetland area") and were adapted for freshwater fish habitats due to the lack of species-specific protective benchmarks for endemic desert fishes. Although there are 3 designated Ramsar sites in Oman, comprehensive data on Oman's total wetland area are not publicly available, which limits our ability to understand these wetland conservation rates as part of an overall wetland conservation framework. In addition to this limitation, our gap assessment assumes that all of the 31 protected areas in Oman are managed uniformly, yet the evidence cited demonstrates a variability in management effectiveness and enforcement [247–249],

thus potentially compromising conservation outcomes even within formally designated reserves This gap analysis did not consider the role of private land conservation initiatives, community-managed water resources, nor traditional falaj systems that may serve as de facto protections of freshwater habitats beyond the formal protected area network. Our spatial overlap analysis used the current (2024) protected area boundaries, and therefore does not represent proposed expansions or potential future areas being evaluated for designation as protected by the Environment Authority.

Therefore, future planned management must be treated as possible adaptive management strategies where continuous assessments are sub-routine, there are plans for sampling in under-sampled locations, studies on potential hybridization, and dynamic conservation planning that considers both model uncertainty and acknowledges environmental variability in desert freshwater ecosystems. Regular reassessment of protection gaps as new protected areas are designated and/or as management efficacy increases will be necessary to track the progress toward national and international conservation targets.

## 5. Conclusions

This integrated study provides strong scientific support for the conservation of two endemic/native *Aphaniops* species with high-performance MaxEnt models (AUC 0.974 and 0.950) demonstrating that these species have distinct climate-driven biogeography. The restricted southern range of *A. kruppi* is heavily influenced by monsoon-mediated moisture regimes, while the more widespread but climatically less stable northern range of *A. stoliczkanus* is influenced by winter precipitation. We note species-specific climate change vulnerabilities, indicating the need for different management approaches, especially the increased climate risks associated with non-linear (or extreme) precipitation patterns, and the additional risks to *A. stoliczkanus* associated with relatively rapid anthropocentric change. Our MESS Analysis identified significant climate novelty exposure, with 85–95% of all future projected conditions for *A. kruppi* are expected to exceed the environmental training data currently available for this species; This represents an extraordinary opportunity for proactive conservation efforts to address the unprecedented environmental changes projected into the future.

Our habitat suitability assessment demonstrates an unexpected inverse relationship between the categorizations of optimal habitat and aquatic biodiversity. This trend illustrates the specialization-dominance trade-off whereby *Aphaniops* spp. is able to numerically dominate under harsh environmental conditions by competitively excluding generalist species; this fundamentally undermines traditional conservation notions of habitat quality and biodiversity. These findings raise questions about some conventional assumptions regarding conservation wherein favorable habitat conditions result in greater diversity of species. Furthermore, it is noteworthy that dissolved oxygen became the most important parameter from our research, indicating that when we think about *Aphaniops* species as ecological specialists, we must prioritize habitat considerations for targeted conservation over expanded umbrella conservation. It's important to emphasize that only 25% of sites offer "Highly Suitable" habitat conditions, indicating the rarity and potential conservation value of the optimal conditions present in desert aquatic systems. The NMDS analysis of community composition (stress = 0.044) identified clear spatial segregation among the communities associated with streams; Environmental filtering was shown to be occurring at differential levels across various landscape contexts, including montane-influenced systems (AW sites), traditional wadi networks (K sites), urban gradients (A sites), and urban wadi-to-sea continuum (D sites).

While there is some habitat protection already in operation, with ecological specialists residing within a total area of 17 441.45 km² (31 reserves), our spatial analysis has identified significant issues within the wadi system where these species reside, which can be considered habitat gaps. The current protected area of high suitability habitat for both species remains very low (0.31% for *A. kruppi* and 1.34% for *A. stoliczkanus*), which represent only 4.2% and 17.8% progress towards meeting Oman's national wetland conservation goal of 7.5% by 2040, and also only 1.0%, and 4.5% towards the CBD 30x30 global target. More importantly, nearly 98% of the high-density wadi systems that act as connectivity corridors are unprotected, which would require a 24-fold increase in the area of protected habitat for *A. kruppi*, and a 6-fold increase in the area of protected habitat for *A. stoliczkanus* to meet Oman's national conservation goals. The documented hybridization observed between *A. stoliczkanus* and *A. kruppi* will require a shift from a species-focused conservation regime

 

to a landscape-based regime which represent the connectivity needed for adaptive gene flow, and which could facilitate climate resilience through adaptive introgression.

Our key immediate conservation recommendations include: (1) the creation of multi-habitat wadi corridors that incorporate high and moderate suitability areas with specific density-based protection targets (high-densities = 50%, med-densities = 30%, low-densities = 10%); (2) changes in water quality thresholds that tier dissolved oxygen across habitats (with priority given to maintaining concentrations above 6.42 mg/L in priority habitats); (3) climate-adaptive management strategies that maintain connectivity across *A. kruppi* and *A. stoliczkanus* to maintain populations through climate change while accounting for future habitat shifts under different emission scenarios; and (4) landscape-scale conservation planning that conserves the entire environmental gradient from specialist-dominated to generalist-diverse habitats and protects key protected areas identified in this research (Jabal Samhan Reserve, Al Wusta Wetland Reserve, Khor Kharfour Archaeological Reserve for *A. kruppi*; Al Buraimi Oasis Nature Reserve, Al Serien Nature Reserve, Western Mountain Nature Reserve for *A. stoliczkanus*) for immediate expansion and enhanced management.

This study provides a systematic planning approach that is replicable in arid environments globally, and clearly outlines that protection of specialized desert aquatic fauna will require integrated strategies that are broader than single-species management practices, and consider the compounding impacts of climate change, habitat fragmentation, and anthropogenic disturbance in water limited systems.

## Supporting information

**S1 Fig.** Environmental layers for Oman from WorldClim data.(A) Digital Elevation Model, (B) Mean Annual Temperature, (C) Annual Precipitation.
(TIF)

**S2 Fig.** Species distribution model performance for *Aphaniops kruppi* and *Aphaniops stoliczkanus*.(A) ROC curves with optimal threshold points showing differential predictive performance, and (B) comparison of three threshold methods across seven evaluation metrics.
(TIF)

**S3 Fig.** Predictor variable percent contributions for *Aphaniops* species habitat suitability used in MaxEnt model.
(TIF)

**S4 Fig.** Multivariate Environmental Similarity Surface (MESS) maps for *Aphaniops kruppi.*
(TIF)

**S5 Fig.** Multivariate Environmental Similarity Surface (MESS) maps for *Aphaniops stoliczkanus.*
(TIF)

**S6 Fig.** Predictive uncertainty classification for: (A) *Aphaniops kruppi*, and (B) *Aphaniops stoliczkanus.*
(TIF)

**S7 Fig.** Integrated habitat suitability model performance of *Aphaniops* spp.: (A), (B) metrics, and (C) normality.
(TIF)

**S8 Fig.** Habitat Suitability Index values with 95% bootstrap confidence intervals across 12 stream sites.
(TIF)

**S9 Fig.** Leave-one-out cross-validation results comparing observed and predicted Habitat Suitability Index values.The red dashed line is the perfect prediction line (1:1), while the blue line is the actual regression fit with a 95% Confidence Interval.
(TIF)

**S10 Fig.** Cross-validation residual plot showing prediction errors versus observed Habitat Suitability Index values.The red dashed line is at zero represents perfect prediction, while the blue line with a 95% Confidence Interval shows the systematic pattern in residuals.
(TIF)

**S11 Fig.** Minimal progress toward the 30% global target for native freshwater fishes.Percentage of high-suitability habitat for *Aphaniops kruppi* and *A. stoliczkanus* currently within Oman's protected area network.
(TIF)

**S12 Fig.** Protection status of wadi networks across density classes.Percentage of wadi streams within protected areas, assessed against conservation targets. High-density wadis show a critical gap.
(TIF)

**S1 Table.** Distribution record of *Aphaniops* species in Oman.
(DOCX)

**S2 Table.** Physical and environmental site characteristics in Hajar Mountain wadis, Oman.
(DOCX)

**S3 Table.** Fish abundance and hydrological characteristics of Hajar Mountain wadis, Oman.
(DOCX)

**S4 Table.** Discrimination and calibration metrics for *Aphaniops* species distribution models (SDMs).Calculated from independent test data.
(DOCX)

**S5 Table.** Continuous Boyce Index (CBI) summary statistics for *Aphaniops* species distribution.
(DOCX)

**S6 Table.** Predictive uncertainty summary for *Aphaniops* species distribution models based on 15 bootstrap replicates.
(DOCX)

**S7 Table.** Correlation between Habitat Suitability Index (HSI) and biodiversity metrics with bootstrap confidence intervals.
(DOCX)

**S8 Table.** Bootstrap estimates for Habitat Suitability Index (HSI) and biodiversity metrics.Estimates with 95% confidence intervals from 1,000 iterations across 12 sites.
(DOCX)

**S9 Table.** Leave-One-Out Cross-Validation results for HSI model.
(DOCX)

**S10 Table.** Bray-Curtis dissimilarity matrix of beta diversity among Hajar Mountain stream sites.
(DOCX)

**S11 Table.** Habitat Suitability Curve (HSC) parameter contributions for *Aphaniops* spp.
(DOCX)

**S12 Table.** Optimal values and tolerance ranges from *Aphaniops* spp. Habitat Suitability Curves.
(DOCX)

**S13 Table.** Habitat Suitability Index comparison with and without substrate texture parameter.Analysis across 12 stream sites in Oman's Hajar Mountains.
(DOCX)

**S14 Table.** Site-specific community assembly patterns and environmental drivers in wadi systems.
(DOCX)

## Acknowledgments

We thank the Oman Environment Authority for granting research permits and providing the updated protected areas map through their Geographic Information System (GIS) department. We acknowledge the Environment Authority's National Environment Strategy (2021–2040) and its commitment to evidence-based conservation planning for wetland ecosystems in alignment with Oman Vision 2040. We appreciate the Marine Sciences and Fisheries students at Sultan Qaboos University who helped with field sampling: Humaid Al Mamari, Waheeb Al Wahaibi, Alyasa Al Hasni, Al Hussain Al Hussaini, Majed Al Alawi, and Mohammed Al Shehhi. We also thank volunteers Salim Al Busaidi, Aysha Al Khulifeen, Sara Al Tuwaiya and Asaad Al Alawi for their assistance with field sampling. We would like to thank Dr. Abdallah Akintola for his exceptional efforts and commitment to helping us develop the R code for this research.

## Author contributions

**Conceptualization:** Aziza S Al Adhoobi, Amna Al Ruheili, Saud M. Al Jufaili.

**Data curation:** Aziza S Al Adhoobi, Amna Al Ruheili.

**Formal analysis:** Aziza S Al Adhoobi, Saud M. Al Jufaili.

**Funding acquisition:** Aziza S Al Adhoobi, Saud M. Al Jufaili.

**Investigation:** Aziza S Al Adhoobi.

**Methodology:** Aziza S Al Adhoobi, Amna Al Ruheili, Saud M. Al Jufaili, Wenresti Gallardo.

**Project administration:** Saud M. Al Jufaili.

**Resources:** Aziza S Al Adhoobi, Saud M. Al Jufaili.

**Software:** Aziza S Al Adhoobi, Amna Al Ruheili.

**Supervision:** Amna Al Ruheili, Saud M. Al Jufaili.

**Validation:** Aziza S Al Adhoobi, Amna Al Ruheili, Saud M. Al Jufaili, Wenresti Gallardo.

**Visualization:** Aziza S Al Adhoobi, Amna Al Ruheili.

**Writing – original draft:** Aziza S Al Adhoobi.

**Writing – review & editing:** Aziza S Al Adhoobi, Amna Al Ruheili, Saud M. Al Jufaili, Wenresti Gallardo.

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
