## [Decision Letter · Decision Letter 0]

20 Oct 2025

PONE-D-25-39513Integrated approach to model distribution and assess habitat suitability of killifish species in Oman’s local streams (wadis) under current and future climate conditionsPLOS ONE

Dear Dr. Al Adhoobi,

Thank you for submitting your manuscript to PLOS ONE. After careful consideration, we feel that it has merit but does not fully meet PLOS ONE’s publication criteria as it currently stands. Therefore, we invite you to submit a revised version of the manuscript that addresses the points raised during the review process.

We look forward to receiving your revised manuscript.

Kind regards,

Daniel de Paiva Silva, Ph.D.

Academic Editor

PLOS ONE

Journal Requirements:

“This research was partially funded by the Sultan Qaboos University Financial Support for PhD Students Research Projects and by Sultan Qaboos University under the project number IG/AGR/FISH/22/01.”

“This research was partially funded by the Sultan Qaboos University Financial Support for PhD Students Research Projects and by Sultan Qaboos University under the project number IG/AGR/FISH/22/01.”

4. We note that there is identifying data in the Supporting Information file <S1 Table>. Due to the inclusion of these potentially identifying data, we have removed this file from your file inventory. Prior to sharing human research participant data, authors should consult with an ethics committee to ensure data are shared in accordance with participant consent and all applicable local laws.

-Location data

5. We note that Figures 1-4, 11 & 15 in your submission contain [map/satellite] images which may be copyrighted. All PLOS content is published under the Creative Commons Attribution License (CC BY 4.0), which means that the manuscript, images, and Supporting Information files will be freely available online, and any third party is permitted to access, download, copy, distribute, and use these materials in any way, even commercially, with proper attribution. For these reasons, we cannot publish previously copyrighted maps or satellite images created using proprietary data, such as Google software (Google Maps, Street View, and Earth). For more information, see our copyright guidelines: http://journals.plos.org/plosone/s/licenses-and-copyright.

a. You may seek permission from the original copyright holder of 1-4, 11 & 15 to publish the content specifically under the CC BY 4.0 license.

Additional Editor Comments (if provided):

Dear Dr. Al Adhoobi,

After this first review round, the reviewers indicated positive features of your manuscript that may allow it to be accepted for publication in PLoS One. Still, there are major issues that need to be considered before your study is accepted for publication in this journal. Please consider the suggestions made by the reviewers and prepare a newe version fo the text, along with a rebuttal letter explaining which of the suggestions were included or excluded from the new version of the study.

Sincerely,

Daniel Silva

Reviewers' comments:

Reviewer's Responses to Questions

**Comments to the Author**

1. Is the manuscript technically sound, and do the data support the conclusions?

Reviewer #1: Yes

Reviewer #2: Partly

2. Has the statistical analysis been performed appropriately and rigorously? 

Reviewer #1: Yes

Reviewer #2: Yes

3. Have the authors made all data underlying the findings in their manuscript fully available?

Reviewer #1: Yes

Reviewer #2: Yes

4. Is the manuscript presented in an intelligible fashion and written in standard English?

Reviewer #1: Yes

Reviewer #2: Yes

5. Review Comments to the Author

Reviewer #1: The manuscript addresses an important and timely topic by forecasting habitat suitability for two arid-land killifishes (Aphaniops kruppi and A. stoliczkanus) and linking these results to conservation planning for Omani wadis. The integration of SDMs, habitat suitability indices, community data, and protected-area overlays is promising and highlights the potential for valuable contributions to conservation science. To further strengthen the study, I suggest the following points of improvement:

Model calibration – Provide more detail on how the accessible area (M) was defined, how background points were selected, and whether any bias correction or model tuning procedures were applied.

Performance and thresholds – In addition to discrimination metrics (AUC, TSS), consider reporting calibration metrics (e.g., Boyce, CBI) and briefly discuss limitations related to prevalence mismatches. Clarify the rationale for using equal suitability classes and how thresholding decisions affect interpretation.

Predictor variables – Clarify how predictors were screened for collinearity (e.g., correlation thresholds, VIF), and provide more detail about the source and processing of the elevation data.

Sample size and spatial structure – Acknowledge the relatively small sample size for A. kruppi and specify whether spatial blocking or other methods were used to reduce spatial autocorrelation during model evaluation.

Future projections – Indicate which climate models (ensemble or single GCMs) were used for future scenarios, and include uncertainty and extrapolation maps to strengthen interpretation.

HSI and diversity analyses – Present measures of uncertainty (e.g., confidence intervals) for the HSI and diversity analyses to provide a clearer picture of variability.

Conservation guidance – Strengthen the conservation section by quantifying representation within protected areas and suggesting clear targets (e.g., percentage of suitable habitat to be protected).

Figures – Improve figure readability by adding scale bars, north arrows, clear units, and consistent palettes. Label SSPs and time periods directly on the figures, and consider including response curves with confidence intervals.

Overall, this is a valuable study with strong potential to inform conservation planning in arid ecosystems. Addressing these points will enhance the clarity, transparency, and applicability of the results.

Reviewer #2: The study addresses interesting and important questions about spatial distribution patterns of species at broader scales across integrated climate change scenarios and site-based habitat assessments, raising concerns for conservation planning. I would like to congratulate the authors on developing this research, which I consider highly relevant. However, I also have a few comments.

There are some points in the text that need refinement because I got lost with the number of figures presented and some long paragraphs (mainly in the discussion). There are also points where the discussion mixes with the results, making the text repetitive. The results are robust, and I believe the text would benefit from a complete and substantial revision; please see my additional comments below.

Fig 1. Systematic methodology framework: I did not find it cited in the text.

Fig 2; 5; 6 e 9: My suggestion is to include them as Supporting Information. The environmental variable and model performance plots are interesting but, in my opinion, highlight information that is not the main result.

Tables 2–3: Is it essential to have two separate tables? I would suggest just one table for both species. However, this is at the authors' discretion.

Line 202: It is necessary to state what the acronym UAE means. Intuitively, we can conclude it is "United Arab Emirates," but it's best to avoid confusion.

Lines 237: The GBIF search must be referenced. When performing a search, GBIF creates a citation; thus, I suggest you include it in the text.

Lines 282: This section does not clearly state the use of training data (used to calibrate the distribution models) and test data (used to evaluate predictive ability), nor the proportion of each relative to the total occurrences of the two species. I believe this is relevant information for fully understanding the modeling process and for the experiment's replicability.

Line 308: The authors state that they used the MaxSS from the training data. Evaluating the model on the training data is similar to taking a test with the answer key, as it measures the model's fit to the data it has already seen. Generally, this leads to an overestimation of performance. The model can become overly complex, capturing not only the ecological effect but also noise and specific errors from the training set, which can lead to overfitting, making the model validation biased. Therefore, it is imperative to review the model evaluation metrics, as performance should be assessed using a test dataset. In the case of small datasets (few occurrences), cross-validation is the most recommended technique. Always prioritize evaluation with test data (or via cross-validation) to make decisions about the quality and usefulness of your species distribution model. Evaluation on training data is useful for diagnosing problems like overfitting but should not be used as the final performance measure.

If possible, this procedure could also be adopted for the "Habitat Suitability Index (HSI) model for Aphaniops spp."

These were just a few considerations so that the study can soon be published and available to the community as a product that can serve as a warning (climate change) while offering ways to mitigate negative impacts on Oman's local aquatic communities and killifish species populations.

Congratulations on the manuscript.

6. PLOS authors have the option to publish the peer review history of their article (what does this mean?). If published, this will include your full peer review and any attached files.

Reviewer #1: **Yes:** Luciano Montag

Reviewer #2: No

---

## [Author Response · Author response to Decision Letter 1]

10 Dec 2025

Manuscript ID: PONE-D-25-39513

Title: Integrated approach to model distribution and assess habitat suitability of killifish species in Oman's local streams (wadis) under current and future climate conditions

Journal: PLOS ONE

Response to academic editor

Dear Dr. Daniel de Paiva Silva,

Thank you for the opportunity to revise our manuscript and for the constructive feedback from the reviewers. We have carefully addressed all comments raised during the review process. Below, we provide detailed point-by-point responses to both the journal requirements and reviewer comments, along with descriptions of the changes made to the manuscript.

Below is a summary of the key actions taken to comply with journal requirements and strengthen the manuscript:

1. PLOS ONE style & formatting: The manuscript has been thoroughly reformatted to comply with all PLOS ONE style requirements, including file naming, heading hierarchy, and citation formatting. Figure and table formatting according to journal guidelines. All figures have been regenerated at 300 dpi TIFF format (600 dpi for multi-panel figures). All submitted files now follow the naming conventions: "Manuscript.docx", "Revised Manuscript with Track Changes.docx", and "Response to Reviewers.docx".

2. Funding statements: The complete amended funding statement and a declaration of the funders' role have been provided in the submitted cover letter.

3. Data anonymization: All specific geographic coordinates (longitude and latitude) have been permanently removed from the Supporting Information S1 Table to protect sensitive location data of endemic species and comply with data sharing policies.

4. Figure copyright compliance: All map/satellite imagery has been verified for copyright compliance. Figures 3 and 4 (now Figs 2 and 3) were revised by replacing Esri basemaps with public-domain vector maps from Natural Earth. We have provided a detailed Content Permission letter (submitted as "Other" file) documenting the copyright status of all figures.

5. Citation of recommended works: We confirm that the reviewers' comments did not include any recommendations to cite specific previously published works. However, we have added several methodologically important references in response to reviewers' requests for clarification of our analytical approaches (see detailed responses to Reviewer #1 and #2 below). All new citations were independently evaluated for relevance and appropriateness before inclusion.

6. Substantive revisions: In response to the reviewers’ insightful comments, we have made significant enhancements to the manuscript’s scientific rigor and clarity. Key improvements include:

• Methodological transparency: Detailed descriptions of species distribution model calibration, accessible area (M) definition, background point selection, collinearity screening, and spatial filtering have been added.

• Robust validation: We expanded model evaluation to include calibration metrics (Continuous Boyce Index) and a comprehensive uncertainty framework, including a Multivariate Environmental Similarity Surface (MESS) analysis to assess extrapolation risk in future projections.

• Quantified conservation framework: A major new conservation gap analysis was conducted, with results consolidated in a new table (Table 5). This provides clear, quantified targets against national (Oman Vision 2040) and global (CBD 30x30) benchmarks, identifying specific protection shortfalls and priority actions.

• Improved readability: The discussion was restructured, long paragraphs were shortened, and interpretive statements were removed from the results. Four secondary figures were moved to Supporting Information to streamline the main narrative.

We are confident that these comprehensive revisions have fully addressed all editorial and reviewer concerns. The manuscript is now substantially stronger in methodological rigor, analytical depth, and conservation applicability.

Sincerely,

Aziza S. Al Adhoobi, Ph.D. Candidate

Department of Marine Science and Fisheries, College of Agricultural & Marine Sciences, Sultan Qaboos University, Muscat, Oman

Department of Nature Reserves, Directorate General of Biodiversity and Nature Reserves, Environment Authority, Muscat, Oman

Email: s33756@student.squ.edu.om, azizco83@gmail.com

ORCID: https://orcid.org/0000-0002-7326-8011

On behalf of all co-authors:

• Saud M. Al Jufaili, Sultan Qaboos University

• Amna Al Ruheili, Sultan Qaboos University

• Wenresti Gallardo, Sultan Qaboos University

Response to reviewers

Dear Reviewers,

Thank you for your valuable feedback, which has significantly strengthened our manuscript. We have carefully addressed all comments. Below is a point-by-point summary of the key changes made.

Response to reviewer #1:

Reviewer #1: The manuscript addresses an important and timely topic by forecasting habitat suitability for two arid-land killifishes (Aphaniops kruppi and A. stoliczkanus) and linking these results to conservation planning for Omani wadis. The integration of SDMs, habitat suitability indices, community data, and protected-area overlays is promising and highlights the potential for valuable contributions to conservation science. To further strengthen the study, I suggest the following points of improvement:

Author Response: We sincerely thank the reviewer for recognizing the value and integrative nature of our study. We have carefully addressed all suggestions to strengthen the manuscript's methodological rigor, transparency, and conservation applicability. We have carefully addressed all suggestions.

Reviewer comment #1.1: Model calibration – Provide more detail on how the accessible area (M) was defined, how background points were selected, and whether any bias correction or model tuning procedures were applied.

Author Response: We apologize for the insufficient methodological detail in our original submission. We have substantially revised Methods Sections 2.1.2 and 2.1.4 to include:

Key additions:

• Definition of accessible area (M) using FEOW ecoregions (IDs 443 & 439).

• Rationale for 10,000 background points restricted to M.

• Application of 1 km spatial filtering to reduce autocorrelation.

• Description of model tuning settings (regularization, bootstrap replicates).

Changes made:

Methods Section 2.1.2 "Species occurrence data":

• Lines 245-251: "To reduce spatial autocorrelation (clustering bias) and balance data quality with sample size, we applied 1 km spatial filtering, ..."

• Lines 253-256: "Due to the low number of occurrences of A. kruppi, we decided against using spatial blocking for model evaluation to ensure sufficient data for cross-validation."

• Lines 257-261: "We determined both species' accessible area (M) by the FEOW ecoregion [59] for the ID 443 and 439, which represent the geographical area ..."

Methods Section 2.1.4 "Species distribution modeling with MaxEnt":

• Lines 326-332: "We used MaxEnt default settings, to generate 10,000 random background data points that are spatially limited to the entire defined accessible area (M) where a species can potentially be present and utilize environmental resources available at each location... A 10,000-point background sample was sufficient for describing all potential environmental characteristics on M, while remaining computationally efficient [95]."

Reviewer comment #1.2: In addition to discrimination metrics (AUC, TSS), consider reporting calibration metrics (e.g., Boyce, CBI) and briefly discuss limitations related to prevalence mismatches. Clarify the rationale for using equal suitability classes and how thresholding decisions affect interpretation.

Author Response: Thank you for this important suggestion. We have added calibration metrics and expanded our discussion of limitations:

Changes made:

Methods Section 2.1.5 "Model validation": Added Boyce Index procedure.

• Lines 382-397: "In addition to the threshold dependent metrics mentioned above, we examined the model's calibration using an assessment approach consistent with the Continuous Boyce Index (CBI) to evaluate models' calibration... This method calculates a normalized performance ratio, defined as (mean presence/mean background -- 1)/ (mean presence/mean background + 1)..."

Results Section 3.1.1 "Species distribution model performance":

• Lines 555–561: "The CBI results showed that the models performed exceptionally well for both species: with values of 0.909 ± 0.025 for A. kruppi and 0.872 ± 0.028 for A. stoliczkanus (S4 Table)."

Prevalence mismatches: Discussed in Section 3.1.1 (lines 552–554), Section 4.1 (lines 847-855), and Section 4.4 (lines 1224–1230).

Threshold rationale: Explained in Methods Section 2.1.5 (lines 372–375), and Section 4.1 (lines 919-925).

Reviewer comment #1.3: Clarify how predictors were screened for collinearity (e.g., correlation thresholds, VIF), and provide more detail about the source and processing of the elevation data.

Author Response: We thank the reviewer for this important methodological clarification request. We revised Section 2.1.3 to clarify:

Collinearity screening: “We applied a Pearson Correlation Matrix (threshold: |r| > 0.7). We selected pairwise correlation over Variance Inflation Factor (VIF)…” (lines 282–294)

DEM processing: Details on DEM source and derived metrics (lines 295–301).

Changes made: Section 2.1.3 (lines 282–301) comprehensively documents these protocols.

Reviewer comment #1.4: Acknowledge the relatively small sample size for A. kruppi and specify whether spatial blocking or other methods were used to reduce spatial autocorrelation during model evaluation.

Author Response: We have explicitly acknowledged the sample size limitation for A. kruppi and clarified our spatial filtering approach. The 1km spatial filtering was implemented to reduce spatial autocorrelation, and we have explained why spatial blocking was not feasible given the sample size constraints inherent to studying endemic species.

Changes made:

• Methods Section 2.1.2 (lines 245–252): Enhanced description of spatial filtering.

• Discussion Section 4.1 (lines 999–1004): Added paragraph on sample size implications.

Reviewer comment #1.5: Indicate which climate models (ensemble or single GCMs) were used for future scenarios, and include uncertainty and extrapolation maps to strengthen interpretation.

Author Response: We thank the reviewer for this important clarification. We specified the use of and added a comprehensive uncertainty framework:

To comprehensively address uncertainty concerns, we implemented a three-pronged approach:

• Bootstrap uncertainty quantification across 15 model replicates to assess model stability

• Multivariate Environmental Similarity Surface (MESS) analysis to identify extrapolation risk areas (new Section 2.1.6)

• Comprehensive model evaluation using independent test data and multiple validation metrics

Changes made:

• GFDL-ESM4 GCM: Section 2.1.3 (lines 269-281)

• MESS analysis: New Section 2.1.6 (lines 399–413)

• Results: New Section 3.1.4 (lines 626–644)

• Discussion: Section 4.1.3 (lines 921–935) discuss extrapolation risk.

• Supporting materials: Table 3, S4–S6 Figs, and S6 Table.

Reviewer comment #1.6: Present measures of uncertainty (e.g., confidence intervals) for the HSI and diversity analyses to provide a clearer picture of variability.

Author Response: We thank the reviewer for this important suggestion. We have incorporated comprehensive uncertainty measures through bootstrap resampling (1,000 iterations), generating 95% confidence intervals for all correlation coefficients between HSI and biodiversity metrics.

Changes made:

• Methods Section 2.2.2 (lines 487-490): "To address potential overfitting, and quantify uncertainty, we utilized bootstrap resampling (1000 iterations), to create 95 % confidence intervals for correlation coefficients and Leave-One-Out Cross-Validation (LOOCV)… "

• Results Section 3.2.2 (lines 663-679)

• Supporting materials: Table 4, Fig 6, S Tables 7-9, and S Figs 7-10

Reviewer comment #1.7: Strengthen the conservation section by quantifying representation within protected areas and suggesting clear targets (e.g., percentage of suitable habitat to be protected).

Author Response: We sincerely thank the reviewer for their insightful and constructive comments, which have significantly strengthened our manuscript. We have conducted a comprehensive conservation gap analysis and revised the manuscript accordingly.

Changes made:

• Methods Sections 2.3.1 and 2.3.2 (lines 505-531)

• Result Section 3.3 (lines 792-820)

• Discussion Section 4.3: Section 4.3.1 (lines 1180-1191), Section 4.3.2 (lines 1192-1200), and Section 4.3.3 (lines 1201-1218).

• Supporting materials: New Table 5, Figs 11 and 12, and S Figs 11 and 12.

Reviewer comment #1.8: Improve figure readability by adding scale bars, north arrows, clear units, and consistent palettes. Label SSPs and time periods directly on the figures.

Author Response: We have revised all figures accordingly. All figures have been regenerated at 300 dpi TIFF format (600 dpi for multi-panel figures) for publication quality and uploaded to the Preflight Analysis and Conversion Engine (PACE) digital diagnostic tool to ensure that figures meet PLOS requirements and were checked in NAAS before upload.

Changes made: All figures revised with improved readability elements; figure files resubmitted.

Response to reviewer #2:

Reviewer #2: The study addresses interesting and important questions about spatial distribution patterns of species at broader scales across integrated climate change scenarios and site-based habitat assessments, raising concerns for conservation planning. However, there are some points in the text that need refinement.

Author Response: We sincerely thank the reviewer for the positive assessment and constructive feedback. We have carefully revised the manuscript to address all concerns regarding text organization, figure presentation, and methodological clarity.

Reviewer comment #2.1: There are some points in the text that need refinement because I got lost with the number of figures presented and some long paragraphs (mainly in the discussion). There are also points where the discussion mixes with the results, making the text repetitive.

Author Response: We have substantially revised the manuscript to improve clarity and readability:

Changes made:

• Discussion Sections 4.1–4.3 (lines 838–1218): Shorter paragraphs, clearer organization.

• Added transitional phrases throughout Discussion for improved coherence

Reviewer comment #2.2: Fig 1. Systematic methodology framework: I did not find it cited in the text.

Author Response: Thank you for catching this omission. Figure 1 is now cited in:

• Introduction (line 172): “Our research follows a comprehensive systematic workflow and has three interrelated aims (Fig 1).”

• Methods overview (line 192): “This study follows a comprehensive systematic workflow, illustrated in Fig 1.”

Reviewer comment #2.3: My suggestion is to include them as Supporting Information. The environmental variable and model performance plots are interesting but, in my opinion, highlight information that is not the main result.

Author Response: Thank you for this excellent suggestion. We have moved these figures to Supporting Information to streamline the main text and focus on primary conservation findings:

• Figure 2 (Digital elevation, temperature, precipitation maps) → Supplementary S1 Fig

• Figure 5 (ROC curves and threshold comparison) → Supplementary S2 Fig

• Figure 6 (Predictor variable contributions) → Supplementary S3 Fig

• Figure 9 (HSI model performance metrics) → Supplementary S7 Fig

Reviewer comment #2.4: Is it essential to have two separate tables? I would suggest just one table for both species.

Author Response: Thank you for this suggestion. We have merged Tables 2 and 3 into a single comprehensive table (new Table 2) that facilitates direct comparison of habitat projections for both species across all scenarios and time periods.

Reviewer comment #2.5: It is necessary to state what the acronym UAE means.

Author Response: Thank you for this catch. We have corrected line 210 to read:

" The Oman Mountain ecoregion with the designa

---

## [Decision Letter · Decision Letter 1]

22 Mar 2026

Integrated approach to model distribution and assess habitat suitability of killifish species in Oman's local streams (wadis) under current and future climate conditions

PONE-D-25-39513R1

Dear Dr. Al Adhoobi,

We’re pleased to inform you that your manuscript has been judged scientifically suitable for publication and will be formally accepted for publication once it meets all outstanding technical requirements.

Kind regards,

Daniel de Paiva Silva, Ph.D.

Academic Editor

PLOS One

Additional Editor Comments (optional):

Reviewers' comments:

Reviewer's Responses to Questions

**Comments to the Author**

1. If the authors have adequately addressed your comments raised in a previous round of review and you feel that this manuscript is now acceptable for publication, you may indicate that here to bypass the “Comments to the Author” section, enter your conflict of interest statement in the “Confidential to Editor” section, and submit your "Accept" recommendation.

Reviewer #2: All comments have been addressed

Reviewer #3: All comments have been addressed

2. Is the manuscript technically sound, and do the data support the conclusions?

Reviewer #2: Yes

Reviewer #3: Yes

3. Has the statistical analysis been performed appropriately and rigorously? 

Reviewer #2: Yes

Reviewer #3: Yes

4. Have the authors made all data underlying the findings in their manuscript fully available?

Reviewer #2: Yes

Reviewer #3: Yes

5. Is the manuscript presented in an intelligible fashion and written in standard English?

Reviewer #2: Yes

Reviewer #3: Yes

6. Review Comments to the Author

Reviewer #2: I thank the authors for their dedicated attention and for the careful and comprehensive implementation of the suggestions presented in the previous round of review; the changes made have fully addressed the issues raised, substantially enhancing the clarity, methodological rigor, and overall impact of the manuscript.

Reviewer #3: I would like to congratulate the authors on their excellent revision of the manuscript, as the detailed analysis of the updated version and the response letter to the reviewers demonstrates that all previous concerns have been addressed with due scientific rigor. The study presents an integrated and innovative approach by combining species distribution modeling with the field habitat suitability index, resulting in a robust tool for conservation in arid ecosystems. The inclusion of additional validation metrics, such as the Continuous Boyce Index and uncertainty analysis, significantly strengthened the credibility of the climate projections for the year 2100, while the application of the Jackknife method was a technically appropriate response to address the limited sampling of the species Aphaniops kruppi. Furthermore, linking the results to the goals of Oman Vision 2040 and global biodiversity commitments elevates the manuscript from a theoretical study to a practical and timely high-impact contribution. The text now features a cohesive narrative, a balanced discussion of its limitations, and grammatically correct flow, with no outstanding ethical concerns. Given the methodological maturity achieved and the relevance of the findings for water resource management and biodiversity conservation, I recommend accepting this manuscript for publication.

7. PLOS authors have the option to publish the peer review history of their article (what does this mean?). If published, this will include your full peer review and any attached files.

Reviewer #2: No

Reviewer #3: **Yes:** Emilly Layne Martins do Nascimento

---

## [Editor Report · Acceptance letter]

PONE-D-25-39513R1

PLOS One

Dear Dr. Al Adhoobi,

I'm pleased to inform you that your manuscript has been deemed suitable for publication in PLOS One. Congratulations! Your manuscript is now being handed over to our production team.

Kind regards,

on behalf of

Dr. Daniel de Paiva Silva

Academic Editor

PLOS One